# A Bandit Approach to Multiple Testing with False Discovery Control

**Kevin Jamieson**[*,†], **Lalit Jain**[*]
{jamieson,lalitj}@cs.washington.edu
[*]Paul G. Allen School of Computer Science & Engineering,
University of Washington, Seattle, WA, and
[†]Optimizely, San Francisco, CA

## Abstract

We propose an adaptive sampling approach for multiple testing which aims to maximize statistical power while ensuring anytime false discovery control. We consider $n$ distributions whose means are partitioned by whether they are below or equal to a baseline (nulls), versus above the baseline (actual positives). In addition, each distribution can be sequentially and repeatedly sampled. Inspired by the multi-armed bandit literature, we provide an algorithm that takes as few samples as possible to exceed a target true positive proportion (i.e. proportion of actual positives discovered) while giving anytime control of the false discovery proportion (nulls predicted as actual positives). Our sample complexity results match known information theoretic lower bounds and through simulations we show a substantial performance improvement over uniform sampling and an adaptive elimination style algorithm. Given the simplicity of the approach, and its sample efficiency, the method has promise for wide adoption in the biological sciences, clinical testing for drug discovery, and online A/B/n testing problems.

## 1 Introduction

Consider $n$ possible treatments, say, drugs in a clinical trial, where each treatment either has a positive expected effect relative to a baseline (actual positive), or no difference (null), with a goal of identifying as many actual positive treatments as possible. If evaluating the $i$th trial results in a noisy outcome (e.g. due to variance in the actual measurement or just diversity in the population) then given a total measurement budget of $B$, it is standard practice to execute and average $B/n$ measurements of each treatment, and then output a set of predicted actual positives based on the measured effect sizes. False alarms (i.e. nulls predicted as actual positives) are controlled by either controlling *family-wise error rate (FWER)*, where one bounds the probability that at least one of the predictions is null, or *false discovery rate (FDR)*, where one bounds the expected proportion of the number of predicted nulls to the number of predictions. FDR is a weaker condition than FWER but is often used in favor of FWER because of its higher *statistical power*: more actual positives are output as predictions using the same measurements.

In the pursuit of even greater statistical power, there has recently been increased interest in the biological sciences to reject the uniform allocation strategy of $B/n$ trials to the $n$ treatments in favor of an *adaptive* allocation. Adaptive allocations partition the budget $B$ into sequential rounds of measurements in which the measurements taken at one round inform which measurements are taken in the next [1, 2]. Intuitively, if the effect size is relatively large for some treatment, fewer trials will be necessary to identify that treatment as an actual positive relative to the others, and that savings of measurements can be allocated towards treatments with smaller effect sizes to boost the signal. However, both [1, 2] employed ad-hoc heuristics which may not only have sub-optimal

statistical power, but also may even result in more false alarms than expected. As another example, in the domain of A/B/n testing in online environments, the desire to understand and maximize click-through-rate across treatments (e.g., web-layouts, campaigns, etc.) has become ubiquitous across retail, social media, and headline optimization for the news. And in this domain, the desire for statistically rigorous adaptive sampling methods with high statistical power are explicit [3].

In this paper we propose an adaptive measurement allocation scheme that achieves near-optimal statistical power subject to FWER or FDR false alarm control. Perhaps surprisingly, we show that even if the treatment effect sizes of the actual positives are identical, adaptive measurement allocation can still substantially improve statistical power. That is, more actual positives can be predicted using an adaptive allocation relative to the uniform allocation under the same false alarm control.

## 1.1 Problem Statement

Consider $n$ distributions (or arms) and a game where at each time $t$, the player chooses an arm $i \in [n] := \{1, \ldots, n\}$ and immediately observes a reward $X_{i,t} \overset{iid}{\sim} \nu_i$ where $X_{i,t} \in [0,1]$[1] and $\mathbb{E}_{\nu_i}[X_{i,t}] = \mu_i$. For a *known* threshold $\mu_0$, define the sets[2]

$$\mathcal{H}_1 = \{i \in [n] : \mu_i > \mu_0\} \quad \text{and} \quad \mathcal{H}_0 = \{i \in [n] : \mu_i = \mu_0\} = [n] \setminus \mathcal{H}_1.$$

The value of the means $\mu_i$ for $i \in [n]$ and the cardinality of $\mathcal{H}_1$ are *unknown*. The arms (treatments) in $\mathcal{H}_1$ have means greater than $\mu_0$ (positive effect) while those in $\mathcal{H}_0$ have means equal to $\mu_0$ (no effect over baseline). At each time $t$, after the player plays an arm, she also outputs a set of indices $\mathcal{S}_t \subseteq [n]$ that are interpreted as *discoveries* or rejections of the null-hypothesis (that is, if $i \in \mathcal{S}_t$ then the player believes $i \in \mathcal{H}_1$). For as small a $\tau \in \mathbb{N}$ as possible, the goal is to have the number of true detections $|\mathcal{S}_t \cap \mathcal{H}_1|$ be approximately $|\mathcal{H}_1|$ for all $t \geq \tau$, subject to the number of false alarms $|\mathcal{S}_t \cap \mathcal{H}_0|$ being small uniformly over all times $t \in \mathbb{N}$. We now formally define our notions of false alarm control and true discoveries.

**Definition 1** (False Discovery Rate, FDR-$\delta$). *Fix some $\delta \in (0,1)$. We say an algorithm is FDR-$\delta$ if for all possible problem instances $(\{\nu_i\}_{i=1}^n, \mu_0)$ it satisfies $\mathbb{E}[\frac{|\mathcal{S}_t \cap \mathcal{H}_0|}{|\mathcal{S}_t| \vee 1}] \leq \delta$ for all $t \in \mathbb{N}$ simultaneously.*

**Definition 2** (Family-wise Error Rate, FWER-$\delta$). *Fix some $\delta \in (0,1)$. We say an algorithm is FWER-$\delta$ if for all possible problem instances $(\{\nu_i\}_{i=1}^n, \mu_0)$ it satisfies $\mathbb{P}(\bigcup_{t=1}^{\infty}\{\mathcal{S}_t \cap \mathcal{H}_0 \neq \emptyset\}) \leq \delta$.*

Note FWER-$\delta$ implies FDR-$\delta$, the former being a stronger condition than the latter. Allowing a relatively small number of false discoveries is natural, especially if $|\mathcal{H}_1|$ is relatively large. Because $\mu_0$ is known, there exist schemes that guarantee FDR-$\delta$ or FWER-$\delta$ even if the arm means $\mu_i$ and the cardinality of $\mathcal{H}_1$ are unknown (see Section 2.1). It is also natural to relax the goal of identifying *all* arms in $\mathcal{H}_1$ to simply identifying a *large proportion* of them.

**Definition 3** (True Positive Rate, TPR-$\delta, \tau$). *Fix some $\delta \in (0,1)$. We say an algorithm is TPR-$\delta, \tau$ on an instance $(\{\nu_i\}_{i=1}^n, \mu_0)$ if $\mathbb{E}[\frac{|\mathcal{S}_t \cap \mathcal{H}_1|}{|\mathcal{H}_1|}] \geq 1 - \delta$ for all $t \geq \tau$.*

**Definition 4** (Family-wise Probability of Detection, FWPD-$\delta, \tau$). *Fix some $\delta \in (0,1)$. We say an algorithm is FWPD-$\delta, \tau$ on an instance $(\{\nu_i\}_{i=1}^n, \mu_0)$ if $\mathbb{P}(\mathcal{H}_1 \subseteq \mathcal{S}_t) \geq 1 - \delta$ for all $t \geq \tau$.*

Note that FWPD-$\delta, \tau$ implies TPR-$\delta, \tau$, the former being a stronger condition than the latter. Also note $\mathbb{P}(\bigcup_{t=1}^{\infty}\{\mathcal{S}_t \cap \mathcal{H}_0 \neq \emptyset\}) \leq \delta$ and $\mathbb{P}(\mathcal{H}_1 \subseteq \mathcal{S}_\tau) \geq 1 - \delta$ together imply $\mathbb{P}(\mathcal{H}_1 = \mathcal{S}_\tau) \geq 1 - 2\delta$. We will see that it is possible to control the number of false discoveries $|\mathcal{S}_t \cap \mathcal{H}_0|$ regardless of how the player selects arms to play. It is the rate at which $\mathcal{S}_t$ includes $\mathcal{H}_1$ that can be thought of as the statistical power of the algorithm, which we formalize as its *sample complexity*:

**Definition 5** (Sample Complexity). *Fix some $\delta \in (0,1)$ and an algorithm $\mathcal{A}$ that is FDR-$\delta$ (or FWER-$\delta$) over all possible problem instances. Fix a particular problem instance $(\{\nu_i\}_{i=1}^n, \mu_0)$. At each time $t \in \mathbb{N}$, $\mathcal{A}$ chooses an arm $i \in [n]$ to obtain an observation from, and before proceeding to the next round outputs a set $\mathcal{S}_t \subseteq [n]$. The* sample complexity *of $\mathcal{A}$ on this instance is the smallest time $\tau \in \mathbb{N}$ such that $\mathcal{A}$ is TPR-$\delta, \tau$ (or FWPD-$\delta, \tau$).*

The sample complexity and value of $\tau$ of an algorithm will depend on the particular instance $(\{\nu_i\}_{i=1}^n, \mu_0)$. For example, if $\mathcal{H}_1 = \{i \in [n] : \mu_i = \mu_0 + \Delta\}$ and $\mathcal{H}_0 = [n] \setminus \mathcal{H}_1$, then we expect the

|  | **False alarm control** | |
| **Detection Probability** | FDR-$\delta$ <br> $\max_t \mathbb{E}[\frac{\|\mathcal{S}_t \cap \mathcal{H}_0\|}{\|\mathcal{S}_t\| \vee 1}] \leq \delta$ | FWER-$\delta$ <br> $\mathbb{P}(\bigcup_{t=1}^{\infty}\{\mathcal{S}_t \cap \mathcal{H}_0 \neq \emptyset\}) \leq \delta$ |
| --- | --- | --- |
| TPR-$\delta, \tau$ <br> $\mathbb{E}[\frac{\|\mathcal{S}_\tau \cap \mathcal{H}_1\|}{\|\mathcal{H}_1\|}] \geq 1 - \delta$ | Theorem 2 <br> $n\Delta^{-2}$ | Theorem 5 <br> $(n-k)\Delta^{-2} + k\Delta^{-2}\log(n-k)$ |
| FWPD-$\delta, \tau$ <br> $\mathbb{P}(\mathcal{H}_1 \subseteq \mathcal{S}_\tau) \geq 1 - \delta$ | Theorem 3 <br> $(n-k)\Delta^{-2}\log(k) + k\Delta^{-2}$ | Theorem 4 <br> $(n-k)\Delta^{-2}\log(k) + k\Delta^{-2}\log(n-k)$ |

Table 1: Informal summary of sample complexity results proved in this paper for $|\mathcal{H}_1| = k$, constant $\delta$ (e.g., $\delta = .05$) and $\Delta = \min_{i \in \mathcal{H}_1} \mu_i - \mu_0$. Uniform sampling across all settings requires at least $n\Delta^{-2}\log(n/k)$ samples, and in the FWER+FWPD setting requires $n\Delta^{-2}\log(n)$. Constants and $\log\log$ factors are ignored.

sample complexity to increase as $\Delta$ decreases since at least $\Delta^{-2}$ samples are necessary to determine whether an arm has mean $\mu_0$ versus $\mu_0 + \Delta$. The next section will give explicit cases.

**Remark 1** (Impossibility of stopping time). *We emphasize that just as in the non-adaptive setting, at no time can an algorithm* stop *and declare that it is TPR-$\delta, \tau$ or FWPD-$\delta, \tau$ for any finite $\tau \in \mathbb{N}$. This is because there may be an arm in $\mathcal{H}_1$ with a mean infinitesimally close to $\mu_0$ but distinct such that no algorithm can determine whether it is in $\mathcal{H}_0$ or $\mathcal{H}_1$. Thus, the algorithm must run indefinitely or until it is stopped externally. However, using an anytime confidence bound (see Section 2) one can always make statements like "either $\mathcal{H}_1 \subseteq \mathcal{S}_t$, or $\max_{i \in \mathcal{H}_1 \setminus \mathcal{S}_t} \mu_i - \mu_0 \leq \epsilon$" where the $\epsilon$ will depend on the width of the confidence interval.*

### 1.2 Contributions and Informal Summary of Main Results

In Section 2 we propose an algorithm that handles all four combinations of {FDR-$\delta$, FWER-$\delta$} and {TPR-$\delta, \tau$, FWPD-$\delta, \tau$}. A reader familiar with the multi-armed bandit literature would expect an adaptive sampling algorithm to have a large advantage over uniform sampling when there is a large diversity in the means of $\mathcal{H}_1$ since larger means can be distinguished from $\mu_0$ with fewer samples. However, one should note that to declare all of $\mathcal{H}_1$ as discoveries, one must sample every arm in $\mathcal{H}_0$ *at least* as many times as the *most sampled* arm in $\mathcal{H}_1$, otherwise they are statistically indistinguishable. As discoveries are typically uncovering rare phenomenon, it is common to assume $|\mathcal{H}_1| = n^\beta$ for $\beta \in (0, 1)$ [4, 5], or $|\mathcal{H}_1| = o(n)$, but this implies that the number of samples taken from the arms in $\mathcal{H}_1$, regardless of how samples are allocated to those arms, will almost always be dwarfed by the number of samples allocated to those arms in $\mathcal{H}_0$ since there are $\Omega(n)$ of them. This line of reasoning, in part, is what motivates us to give our sample complexity results in terms of the quantities that best describe the contributions from those arms in $\mathcal{H}_0$, namely, the cardinality $|\mathcal{H}_1| = n - |\mathcal{H}_0|$, the confidence parameter $\delta$ (e.g., $\delta = .05$), and the gap $\Delta := \min_{i \in \mathcal{H}_1} \mu_i - \mu_0$ between the means of the arms in $\mathcal{H}_0$ and the smallest mean in $\mathcal{H}_1$. Reporting sample complexity results in terms of $\Delta$ also allows us to compare to known lower bounds in the literature [6, 4, 7, 8]. Nevertheless, we do address the case where the means of $\mathcal{H}_1$ are varied in Theorem 2.

An informal summary of the sample complexity results proven in this work are found in Table 1 for $|\mathcal{H}_1| = k$. For the least strict setting of FDR+TPR, the upper-left quadrant of Table 1 matches the lower bound of [4], a sample complexity of just $\Delta^{-2}n$. In this FDR+TPR setting (which requires the fewest samples of the four settings), uniform sampling which pulls each arm an equal number of times has a sample complexity of at least $n\Delta^{-2}\log(n/|\mathcal{H}_1|)$ (see Theorem 7 in Appendix G), which exceeds all results in Table 1 demonstrating the statistical power gained by adaptive sampling. For the most strict setting of FWER+FWPD, the lower-right quadrant of Table 1 matches the lower bounds of [7, 9, 8], a sample complexity of $(n-k)\Delta^{-2}\log(k) + k\Delta^{-2}\log(n-k)$. Uniform sampling in the FWER+FWPD setting has a sample complexity lower bounded by $n\Delta^{-2}\log(n)$ (see Theorem 8 in Appendix G). The settings of FDR+FWPD and FWER+TPR are sandwiched between these results, and we are unaware of existing lower bounds for these settings.

All the results in Table 1 are novel, and to the best of our knowledge are the first non-trivial sample complexity results for an adaptive algorithm in the *fixed confidence* setting where a desired confidence $\delta$ is set, and the algorithm attempts to minimize the number of samples taken to meet the desired conditions. We also derive tools that we believe may be useful outside this work: for always valid $p$-values (c.f. [3, 10]) we show that FDR is controlled for all times using the Benjamini-Hochberg

procedure [11] (see Lemma 1), and also provide an anytime high probability bound on the false discovery proportion (see Lemma 2).

Finally, as a direct consequence of the theoretical guarantees proven in this work and the empirical performance of the FDR+TPR variant of the algorithm on real data, an algorithm faithful to the theory was implemented and is in use in production at a leading A/B testing platform [12].

## 1.3 Related work

Identifying arms with means above a threshold, or equivalently, multiple testing via rejecting null-hypotheses with small $p$-values, is an ubiquitous problem in the biological sciences. In the standard setup, each arm is given an equal number of measurements (i.e., a uniform sampling strategy), a $p$-value $P_i$ is produced for each arm where $\mathbb{P}(P_i \leq x) \leq x$ for all $x \in (0,1]$ and $i \in \mathcal{H}_0$, and a procedure is then run on these $p$-values to declare small $p$-values as rejections of the null-hypothesis, or discoveries. For a set of $p$-values $P_1 \leq P_2 \leq \cdots \leq P_n$, the so-called Bonferroni selection rule selects $\mathcal{S}_{BF} = \{i : P_i \leq \delta/n\}$. The fact that FWER control implies FDR control, $\mathbb{E}[|\mathcal{S}_{BF} \cap \mathcal{H}_0|] \leq \mathbb{P}(\bigcup_{i \in \mathcal{H}_0}\{P_i \leq \delta/n\}) \leq \delta \frac{|\mathcal{H}_0|}{n} \leq \delta$, suggests that greater statistical power (i.e. more discoveries) could be achieved with procedures designed specifically for FDR. The BH procedure [11] is one such procedure to control FDR and is widely used in practice (with its many extensions [6] and performance investigations [5]). Recall that a uniform measurement strategy where every arm is sampled the same number of times requires $n\Delta^{-2}\log(n/k)$ samples in the FDR+TPR setting, and $n\Delta^{-2}\log(n)$ samples in the FWER+FWPD setting (Theorems 7 and 8 in Appendix G), which can be substantially worse than our adaptive procedure (see Table 1).

Adaptive sequential testing has been previously addressed in the *fixed budget* setting: the procedure takes a sampling budget as input, and the guarantee states that if the given budget is larger than a problem dependent constant, the procedure drives the error probability to zero and the detection probability to one. One of the first methods called *distilled sensing* [13] assumed that arms from $\mathcal{H}_0$ were Gaussian with mean at most $\mu_0$, and successively discarded arms after repeated sampling by thresholding at $\mu_0$–at most the median of the null distribution–thereby discarding about half the nulls at each round. The procedure made guarantees about FDR and TPR, which were later shown to be nearly optimal [4]. Specifically, [4, Corollary 4.2] implies that any procedure with $\max\{FDR + (1 - TPR)\} \leq \delta$ requires a budget of at least $\Delta^{-2}n\log(1/\delta)$, which is consistent with our work. Later, another thresholding algorithm for the fixed budget setting addressed the FWER and FWPD metrics [7]. In particular, if their procedure is given a budget exceeding $(n - |\mathcal{H}_1|)\Delta^{-2}\log(|\mathcal{H}_1|) + |\mathcal{H}_1|\Delta^{-2}\log(n - |\mathcal{H}_1|)$ then the FWER is driven to zero, and the FWPD is driven to one. By appealing to the optimality properties of the SPRT (which knows the distributions precisely) it was argued that this is optimal. These previous works mostly focused on the asymptotic regime as $n \to \infty$ and $|\mathcal{H}_1| = o(n)$.

Our paper, in contrast to these previous works considers the *fixed confidence* setting: the procedure takes a desired FDR (or FWER) and TPR (or FWPD) and aims to minimize the number of samples taken before these constraints are met. To the best of our knowledge, our paper is the first to propose a scheme for this problem in the fixed confidence regime with near-optimal sample complexity guarantees.

A related line of work is the threshold bandit problem, where all the means of $\mathcal{H}_1$ are assumed to be strictly above a given threshold, and the means of $\mathcal{H}_0$ are assumed to be strictly below the threshold [14, 15]. To identify this partition, each arm must be pulled a number of times inversely proportional to the square of its deviation from the threshold. This contrasts with our work, where the majority of arms may have means *equal* to the threshold and the goal is to identify arms with means greater than the threshold subject to discovery constraints. If the arms in $\mathcal{H}_0$ are assumed to be strictly below the threshold it is possible to declare arms as in $\mathcal{H}_0$. In our setting we can only ever determine that an arm is in $\mathcal{H}_1$ and not $\mathcal{H}_0$, but it is impossible to detect that an arm is in $\mathcal{H}_0$ and not in $\mathcal{H}_1$.

Note that the problem considered in this paper is very related to the top-$k$ identification problem where the objective is to identify the unique $k$ arms with the highest means with high probability [16, 9, 8]. Indeed, if we knew $|\mathcal{H}_1|$, then our FWER+FWPD setting is equivalent to the top-$k$ problem with $k = |\mathcal{H}_1|$. Lower bounds derived for the top-$k$ problem assume the algorithm has knowledge of the values of the means, just not their indices [16, 8]. Thus, these lower bounds also apply to our setting and are what are referenced in Section 1.2.

---

**Algorithm 1** An algorithm for identifying arms with means above a threshold $\mu_0$ using as few samples as possible subject to false alarm and true discovery conditions. The set $\mathcal{S}_t$ is designed to control FDR at level $\delta$. The set $\mathcal{R}_t$ is designed to control FWER at level $\delta$.

---

**Input:** Threshold $\mu_0$, confidence $\delta \in (0, e^{-1}]$, confidence interval $\phi(\cdot, \cdot)$
**Initialize:** Pull each arm $i \in [n]$ once and let $T_i(t)$ denote the number of times arm $i$ has been pulled up to time $t$. Set $\mathcal{S}_{n+1} = \emptyset$, $\mathcal{R}_{n+1} = \emptyset$, and
**If** TPR
$\qquad \xi_t = 1, \qquad$ and $\qquad \nu_t = 1 \quad \forall t$
**Else if** FWPD
$\qquad \xi_t = \max\{2|\mathcal{S}_t|, \frac{5}{3(1-4\delta)} \log(1/\delta)\}, \qquad$ and $\qquad \nu_t = \max\{|\mathcal{S}_t|, 1\} \quad \forall t$

**For** $t = n+1, n+2, \ldots$
$\qquad$ **Pull arm** $I_t = \arg \max\limits_{i \in [n] \setminus \mathcal{S}_t} \widehat{\mu}_{i, T_i(t)} + \phi(T_i(t), \frac{\delta}{\xi_t})$,
$\qquad$ **Apply** Benjamini-Hochberg [11] selection at level $\delta' = \frac{\delta}{6.4 \log(36/\delta)}$ to obtain $\delta$ FDR-controlled set $\mathcal{S}_t$:
$$s(k) = \{i \in [n] : \widehat{\mu}_{i, T_i(t)} - \phi(T_i(t), \delta' \tfrac{k}{n}) \geq \mu_0\}, \forall k \in [n]$$
$$\mathcal{S}_{t+1} = s(\widehat{k}) \text{ where } \widehat{k} = \max\{k \in [n] : |s(k)| \geq k\} \text{ (if } \nexists \widehat{k} \text{ set } \mathcal{S}_{t+1} = \mathcal{S}_t)$$
$\qquad$ **If** FWER and $\mathcal{S}_t \neq \emptyset$:
$\qquad\qquad$ **Pull arm** $J_t = \arg \max\limits_{i \in \mathcal{S}_t \setminus \mathcal{R}_t} \widehat{\mu}_{i, T_i(t)} + \phi(T_i(t), \frac{\delta}{\nu_t})$
$\qquad\qquad$ **Apply** Bonferroni-like selection to obtain FWER-controlled set $\mathcal{R}_t$:
$$\chi_t = n - (1 - 2\delta'(1 + 4\delta'))|\mathcal{S}_t| + \tfrac{4(1+4\delta')}{3} \log(5 \log_2(n/\delta')/\delta')$$
$$\mathcal{R}_{t+1} = \mathcal{R}_t \cup \{i \in \mathcal{S}_t : \widehat{\mu}_{i, T_i(t)} - \phi(T_i(t), \tfrac{\delta}{\chi_t}) \geq \mu_0\}$$

---

As pointed out by [14], both our setting and the threshold bandit problem can be posed as a combinatorial bandits problem as studied in [17, 18], but such generality leads to unnecessary $\log$ factors. The techniques used in this work aim to reduce extraneous $\log$ factors, a topic of recent interest in the top-1 and top-$k$ arm identification problem [19, 20, 21, 22, 16, 8]. While these works are most similar to exact identification (FWER+FWPD), there also exist examples of *approximate* top-$k$ where the objective is to find any $k$ means that are each within $\epsilon$ of the best $k$ means [9]. Approximate recovery is also studied in a ranking context with a symmetric difference metric [23] which is more similar to the FDR and TPR setting, but neither this nor that work subsumes one another.

Finally, maximizing the number of discoveries subject to a FDR constraint has been studied in a sequential setting in the context of A/B testing with uniform sampling [3]. This work popularized the concept of an always valid $p$-value that we employ here (see Section 2). The work of [10] controls FDR over a *sequence* of independent bandit problems that each outputs at most one discovery. While [10] shares much of the same vocabulary as this paper, the problem settings are very different.

## 2 Algorithm and Discussion

Throughout, we will assume the existence of an *anytime confidence interval*. Namely, if $\widehat{\mu}_{i,t}$ denotes the empirical mean of the first $t$ bounded i.i.d. rewards in $[0, 1]$ from arm $i$, then for any $\delta \in (0, 1)$ we assume the existence of a function $\phi$ such that for any $\delta$ we have $\mathbb{P}\left(\bigcap_{t=1}^{\infty}\{|\widehat{\mu}_{i,t} - \mu_i| \leq \phi(t, \delta)\}\right) \geq 1 - \delta$. We assume that $\phi(t, \delta)$ is non-increasing in its second argument and that there exists an absolute constant $c_\phi$ such that $\phi(t, \delta) \leq \sqrt{\frac{c_\phi \log(\log_2(2t)/\delta)}{t}}$. It suffices to define $\phi$ with this upper bound with $c_\phi = 4$ but there are much sharper known bounds that should be used in practice (e.g., they may take empirical variance into account), see [21, 24, 25, 26]. Anytime bounds constructed with such a $\phi(t, \delta)$ are known to be tight in the sense that $\mathbb{P}(\bigcup_{t=1}^{\infty}\{|\widehat{\mu}_{i,t} - \mu_i| \geq \phi(t, \delta)\}) \leq \delta$ and that there exists an absolute constant $h \in (0, 1)$ such that $\mathbb{P}(\{|\widehat{\mu}_{i,t} - \mu_i| \geq h\,\phi(t, \delta) \text{ for infinitely many } t \in \mathbb{N}\}) = 1$ by the Law of the Iterated Logarithm [27].

Consider Algorithm 1. Before entering the for loop, time-dependent variables $\xi_t$ and $\nu_t$ are defined that should be updated at each time $t$ for different settings. If just FDR control is desired, the algorithm merely loops over the three lines following the for loop, pulling the arm $I_t$ not in $\mathcal{S}_t$ that

has the highest upper confidence bound; such strategies are common for pure-exploration problems [21, 10]. But if FWER control is desired then at most one additional arm $J_t$ is pulled per round to provide an extra layer of filtering and evidence before an arm is added to $\mathcal{R}_t$. Below we describe the main elements of the algorithm and along the way sketch out the main arguments of the analysis, shedding light on the constants $\xi_t$ and $\nu_t$.

## 2.1 False alarm control

$\mathcal{S}_t$ **is FDR-controlled.** In addition to its use as a confident bound, we can also use $\phi(t, \delta)$ to construct:

$$P_{i,t} := \sup\{\alpha \in (0,1] : \widehat{\mu}_{i,t} - \mu_0 \leq \phi(t, \alpha)\} \leq \log_2(2t) \exp(-t(\widehat{\mu}_{i,t} - \mu_0)^2/c_\phi). \quad (1)$$

Proposition 1 of [10] (and the proof of our Lemma 1) shows that if $i \in \mathcal{H}_0$ so that $\mu_i = \mu_0$ then $P_{i,t}$ is an *anytime, sub-uniformly distributed p-value* in the sense that $\mathbb{P}(\bigcup_{t=1}^\infty \{P_{i,t} \leq x\}) \leq x$. Sequences that have this property are sometimes referred to as *always-valid* p-values [3]. Note that if $i \in \mathcal{H}_1$ so that $\mu_i > \mu_0$, we would intuitively expect the sequence $\{P_{i,t}\}_{t=1}^\infty$ to be point-wise smaller than if $\mu_i = \mu_0$ by the property that $\phi(\cdot, \cdot)$ is non-increasing in its second argument. This leads to the intuitive rule to reject the null-hypothesis (i.e., declare $i \notin \mathcal{H}_0$) for those arms $i \in [n]$ where $P_{i,t}$ is very small. The Benjamini-Hochberg (BH) procedure introduced in [11] proceeds by first sorting the p-values so that $P_{(1),T_{(1)}(t)} \leq P_{(2),T_{(2)}(t)} \leq \cdots \leq P_{(n),T_{(n)}(t)}$, then defines $\widehat{k} = \max\{k : P_{(k),T_{(k)}(t)} \leq \delta \frac{k}{n}\}$, and sets $\mathcal{S}_{BH} = \{i : P_{i,T_i(t)} \leq \delta \frac{\widehat{k}}{n}\}$. Note that this procedure is identical to defining sets

$$s(k) = \{i : P_{i,T_i(t)} \leq \delta \tfrac{k}{n}\} = \{i : \widehat{\mu}_{i,T_i(t)} - \phi(T_i(t), \delta \tfrac{k}{n}) \geq \mu_0\},$$

setting $\widehat{k} = \max\{k : |s(k)| \geq k\}$, and $\mathcal{S}_{BH} = s(\widehat{k})$, which is exactly the set $\mathcal{S}_t = \mathcal{S}_{BH}$ in Algorithm 1. Thus, $\mathcal{S}_t$ in Algorithm 1 is equivalent to applying the BH procedure at a level $O(\delta/\log(1/\delta))$ to the anytime p-values of (1). We now discuss the extra logarithmic factor.

Because the algorithm is pulling arms sequentially, some dependence between the p-values may be introduced. Because the anytime p-values are not independent, the BH procedure at level $\delta$ does not directly guarantee FDR-control at level $\delta$. However, it has been shown [28] that for even arbitrarily dependent p-values the BH procedure at level $\delta$ controls FDR at level $\delta \log(n)$ (and that it is nearly tight). Similarly, the following theorem, which may be of independent interest, is a significant improvement when applied to our setting.

**Theorem 1.** *Fix $\delta \in (0, e^{-1})$. Let $p_1, \ldots, p_n$ be random variables such that $\{p_i\}_{i \in \mathcal{H}_0}$ are independent and sub-uniformly distributed so that $\max_{i \in \mathcal{H}_0} \mathbb{P}(p_i \leq x) \leq x$. For any $k \in \{0, 1, \ldots, n\}$, let $R_k := \{i : p_i \leq \delta \frac{k}{n}\}$ and $\widehat{FDP}(R_k) := \frac{\max_{p_i \in R_k} p_i}{|R_k| \vee 1}$.*

$$\mathbb{E}\left[\max_{k:\widehat{FDP}(R_k) \leq \delta} FDP(R_k)\right] \leq \frac{|\mathcal{H}_0|\delta}{n}\left(2\log(\tfrac{2n}{|\mathcal{H}_0|\delta}) + \log(8e^5 \log(\tfrac{8n}{|\mathcal{H}_0|\delta}))\right)$$

$$\leq 4\delta \log(9/\delta)$$

*In other words, any procedure that chooses a set $\{i : p_i \leq \frac{\delta k}{n}\}$ satisfying $|\{i : p_i \leq \frac{\delta k}{n}\}| \geq k$ is FDR controlled at level $O(\delta \log(1/\delta))$.*

Recall, if $\widehat{k} = \max\{k : \widehat{FDP}(R_k) \leq \delta\}$ then $\mathbb{E}[FDP(R_{\widehat{k}})] \leq \delta$ by the standard BH result. When running the algorithm we recommend using BH at level $\delta$, not level $O(\delta/\log(1/\delta))$. As $T_i$ gets very large, $P_{i,T_i(t)} \to P_{i,*}$ and we know that if BH is run on $P_{i,*}$ at level $\delta$ then FDR would be controlled at level $\delta$. We believe this inflation to be somewhat of an artifact of our proofs.

$\mathcal{R}_t$ **is FWER-controlled.** A core obstacle in our analysis is the fact that we don't know the cardinality of $\mathcal{H}_1$. If we did know $|\mathcal{H}_1|$ (and equivalently know $|\mathcal{H}_0| = n - |\mathcal{H}_1|$) then a FWER+FWPD algorithm is equivalent to the so-called top-$k$ multi-armed bandit problem [9, 8] and controlling FWER would be relatively simple using a Bonferroni correction:

$$\mathbb{P}\left(\bigcup_{i \in \mathcal{H}_0} \cup_{t=1}^\infty \{\widehat{\mu}_{i,t} - \phi(t, \tfrac{\delta}{n-|\mathcal{H}_1|}) \geq \mu_0\}\right) \leq \sum_{i \in \mathcal{H}_0} \mathbb{P}\left(\cup_{t=1}^\infty \{\widehat{\mu}_{i,t} - \phi(t, \tfrac{\delta}{|\mathcal{H}_0|}) \geq \mu_0\}\right) \leq |\mathcal{H}_0|\tfrac{\delta}{|\mathcal{H}_0|}$$

which implies FWER-$\delta$. Comparing the first expression immediately above to the definition of $\mathcal{R}_t$ in the algorithm, it is clear our strategy is to use $|\mathcal{S}_t|$ as a surrogate for $|\mathcal{H}_1|$. Note that we could

use the bound $|\mathcal{H}_0| = n - |\mathcal{H}_1| \leq n$ to guarantee FWER-$\delta$, but this could be very loose and induce an $n\log(n)$ sample complexity. Using $|\mathcal{S}_t|$ as a surrogate for $|\mathcal{H}_1|$ in $\mathcal{R}_t$ is intuitive because by the FDR guarantee, we know $|\mathcal{H}_1| \geq \mathbb{E}[|\mathcal{S}_t \cap \mathcal{H}_1|] = \mathbb{E}[|\mathcal{S}_t|] - \mathbb{E}[|\mathcal{S}_t \cap \mathcal{H}_0|] \geq (1-\delta)\mathbb{E}[|\mathcal{S}_t|]$, implying that $|\mathcal{H}_0| = n - |\mathcal{H}_1| \leq n - (1-\delta)\mathbb{E}[|\mathcal{S}_t|]$ which may be much tighter than $n$ if $\mathbb{E}[|\mathcal{S}_t|] \to |\mathcal{H}_1|$. Because we only know $|\mathcal{S}_t|$ and not its expectation, the extra factors in the surrogate expression used in $\mathcal{R}_t$ are used to ensure correctness with high-probability (see Lemma 7).

## 2.2 Sampling strategies to boost statistical power

The above discussion about controlling false alarms for $\mathcal{S}_t$ and $\mathcal{R}_t$ holds for *any* choice of arms $I_t$ and $J_t$ that may be pulled at time $t$. Thus, $I_t$ and $J_t$ are chosen in order to minimize the amount of time necessary to add arms into $\mathcal{S}_t$ and $\mathcal{R}_t$, respectively, and optimize the sample complexity.

**TPR-$\delta, \tau$ setting** implies $\xi_t = \nu_t = 1$. Define the random set $\mathcal{I} = \{i \in \mathcal{H}_1 : \widehat{\mu}_{i,T_i(t)} + \phi(T_i(t), \delta) \geq \mu_i \ \forall t \in \mathbb{N}\}$. Because $\phi$ is an anytime confidence bound, $\mathbb{E}[|\mathcal{I}|] \geq (1-\delta)|\mathcal{H}_1|$. If $\Delta = \min_{i \in \mathcal{H}_1} \mu_i - \mu_0$, then $\min_{i \in \mathcal{I}} \mu_i \geq \mu_0 + \Delta$ and we claim that with probability at least $1 - O(\delta)$ (Section C)

$$\textstyle\sum_{t=1}^{\infty} \mathbf{1}\{I_t \in \mathcal{H}_0, \mathcal{I} \not\subseteq \mathcal{S}_t\} \leq \sum_{t=1}^{\infty} \mathbf{1}\{I_t \in \mathcal{H}_0, \widehat{\mu}_{I_t, T_{I_t}(t)} + \phi(T_{I_t}(t), \delta) \geq \mu_0 + \Delta\}$$
$$\leq c|\mathcal{H}_0|\Delta^{-2}\log(\log(\Delta^{-2}/\delta)).$$

Thus once this number of samples has been taken, either $\mathcal{I} \subseteq \mathcal{S}_t$, or arms in $\mathcal{I}$ will be repeatedly sampled until they are added to $\mathcal{S}_t$ since each arm $i \in \mathcal{I}$ has its upper confidence bound larger than those arms in $\mathcal{H}_0$ by definition. It is clear that an arm in $\mathcal{H}_1$ that is repeatedly sampled will eventually be added to $\mathcal{S}_t$ since its anytime $p$-value of (1) approaches 0 at an exponential rate as it is pulled, and BH selects for low $p$-values. A similar argument holds for $J_t$ and adding arms to $\mathcal{R}_t$.

**Remark 2.** *While the main objective of Algorithm 1 is to identify all arms with means above a given threshold, we note that prior to adding an arm to $\mathcal{S}_t$ in the TPR setting (i.e., when $\xi_t = 1$) Algorithm 1 behaves identically to the nearly optimal best-arm identification algorithm lil'UCB of [21]. Thus, whether the goal is best-arm identification or to identify all arms with means above a certain threshold, Algorithm 1 is applicable.*

**FWPD-$\delta, \tau$ setting** is more delicate and uses inflated values of $\xi_t$ and $\nu_t$. This time, we must ensure that $\{\mathcal{H}_1 \not\subseteq \mathcal{S}_t\} \implies \max_{i \in \mathcal{H}_1 \cap S_t^c} \widehat{\mu}_{i,T_i(t)} + \phi(T_i(t), \delta) \geq \min_{i \in \mathcal{H}_1 \cap S_t^c} \mu_i \geq \mu_0 + \Delta$. Because then we could argue that either $\mathcal{H}_1 \subset \mathcal{S}_t$, or only arms in $\mathcal{H}_1$ are sampled until they are added to $\mathcal{S}_t$ (mirroring the TPR argument). As in the FWER setting above, if we knew the value of $|\mathcal{H}_1|$ the we could set $\xi_t \geq |\mathcal{H}_1|$ to observe that

$$\mathbb{P}(\textstyle\bigcup_{i \in \mathcal{H}_1} \cup_{t=1}^{\infty}\{\widehat{\mu}_{i,t} + \phi(t, \frac{\delta}{\xi_t}) < \mu_i\}) \leq \sum_{i \in \mathcal{H}_1} \mathbb{P}\left(\cup_{t=1}^{\infty}\{\widehat{\mu}_{i,t} + \phi(t, \frac{\delta}{\xi_t}) < \mu_i\}\right) \leq |\mathcal{H}_1|\frac{\delta}{\xi_t}$$

which is less than $\delta$, to guarantee such a condition. But we don't know $|\mathcal{H}_1|$ so we use $|\mathcal{S}_t|$ as a surrogate, resulting in the inflated definitions of $\xi_t$ and $\nu_t$ relative to the TPR setting. The key argument is that either $\mathcal{I} \not\subseteq \mathcal{S}_t$ so that $\max_{i \in \mathcal{I} \cap S_t^c} \widehat{\mu}_{i,T_i(t)} + \phi(T_i(t), \frac{\delta}{\xi_t}) \geq \mu_0 + \Delta$ by the definition of $\mathcal{I}$ (since $\xi_t \geq 1$), or $\mathcal{I} \subset \mathcal{S}_t$ and $|\mathcal{S}_t| \geq \frac{1}{2}|\mathcal{H}_1|$ with high probability which implies $\xi_t = \max\{2|\mathcal{S}_t|, \frac{5}{3(1-4\delta)}\log(1/\delta)\} \geq |\mathcal{H}_1|$ and the union bound of the display above holds.

# 3 Main Results

In what follows, we say $f \lesssim g$ if there exists a $c > 0$ that is independent of all problem parameters and $f \leq cg$. The theorems provide an upper bound on the sample complexity $\tau \in \mathbb{N}$ as defined in Section 1.1 for TPR-$\delta, \tau$ or FWER-$\delta, \tau$ that holds with probability at least $1 - c\delta$ for different values of $c$ [3]. We begin with the least restrictive setting, resulting in the smallest sample complexity of all the results presented in this work. Note the slight generalization in the below theorem where the means of $\mathcal{H}_0$ are assumed to be no greater than $\mu_0$.

**Theorem 2** (FDR, TPR). *Let $\mathcal{H}_1 = \{i \in [n] : \mu_i > \mu_0\}$, $\mathcal{H}_0 = \{i \in [n] : \mu_i \leq \mu_0\}$. Define $\Delta_i = \mu_i - \mu_0$ for $i \in \mathcal{H}_1$, $\Delta = \min_{i \in \mathcal{H}_1} \Delta_i$, and $\Delta_i = \min_{j \in \mathcal{H}_1} \mu_j - \mu_i = \Delta + (\mu_0 - \mu_i)$ for*

$i \in \mathcal{H}_0$. *For all $t \in \mathbb{N}$ we have $\mathbb{E}[\frac{|\mathcal{S}_t \cap \mathcal{H}_0|}{|\mathcal{S}_t| \vee 1}] \leq \delta$. Moreover, with probability at least $1 - 2\delta$ there exists a $T$ such that*

$$T \lesssim \min \left\{ n\Delta^{-2} \log(\log(\Delta^{-2})/\delta), \right.$$
$$\left. \textstyle\sum_{i \in \mathcal{H}_0} \Delta_i^{-2} \log(\log(\Delta_i^{-2})/\delta) + \sum_{i \in \mathcal{H}_1} \Delta_i^{-2} \log(n \log(\Delta_i^{-2})/\delta) \right\}$$

*and $\mathbb{E}[\frac{|\mathcal{S}_t \cap \mathcal{H}_1|}{|\mathcal{H}_1|}] \geq 1 - \delta$ for all $t \geq T$. Neither argument of the minimum follows from the other.*

If the means of $\mathcal{H}_1$ are very diverse so that $\max_{i \in \mathcal{H}_1} \mu_i - \mu_0 \gg \min_{i \in \mathcal{H}_1} \mu_i - \mu_0$ then the second argument of the min in Theorem 2 can be tighter than the first. But as discussed above, this advantage is inconsequential if $|\mathcal{H}_1| = o(n)$. The remaining theorems are given in terms of just $\Delta$. The $\log \log(\Delta^{-2})$ dependence is due to inverting the $\phi$ confidence interval and is unavoidable on at least one arm when $\Delta$ is unknown a priori due to the law of the iterated logarithm [27, 21, 22].

Informally, Theorem 2 states that if just most true detections suffice while not making too many mistakes, then $O(n)$ samples suffice. The first argument of the min is known to be tight in a minimax sense up to doubly logarithmic factors due to the lower bound of [4]. As a consequence of this work, an algorithm inspired by Algorithm 1 in this setting is now in production at one of the largest A/B testing platforms on the web. The full proof of Theorem 2 (and all others) is given in the Appendix due to space.

**Theorem 3** (FDR, FWPD). *For all $t \in \mathbb{N}$ we have $\mathbb{E}[\frac{|\mathcal{S}_t \cap \mathcal{H}_0|}{|\mathcal{S}_t| \vee 1}] \leq \delta$. Moreover, with probability at least $1 - 5\delta$, there exists a $T$ such that*

$$T \lesssim (n - |\mathcal{H}_1|)\Delta^{-2} \log(\max\{|\mathcal{H}_1|, \log\log(n/\delta)\} \log(\Delta^{-2})/\delta) + |\mathcal{H}_1|\Delta^{-2} \log(\log(\Delta^{-2})/\delta)$$

*and $\mathcal{H}_1 \subseteq \mathcal{S}_t$ for all $t \geq T$.*

Here $T$ roughly scales like $(n - |\mathcal{H}_1|) \max\{\log(|\mathcal{H}_1|), \log\log\log(n/\delta)\} + |\mathcal{H}_1|$ where the $\log\log\log(n/\delta)$ term comes from a high probability bound on the false discovery proportion for anytime $p$-values (in contrast to just expectation) in Lemma 2 that may be of independent interest. While negligible for all practical purposes, it appears unnatural and we suspect that this is an artifact of our analysis. We note that if $|\mathcal{H}_1| = \Omega(\log(n))$ then the sample complexity sheds this awkwardness[4].

The next two theorems are concerned with controlling FWER on the set $\mathcal{R}_t$ and determining how long it takes before the claimed detection conditions are satisfied on the set $\mathcal{R}_t$. Note we still have that FDR is controlled on the set $\mathcal{S}_t$ but now this set feeds into $\mathcal{R}_t$.

**Theorem 4** (FWER, FWPD). *For all $t$ we have $\mathbb{E}[\frac{|\mathcal{S}_t \cap \mathcal{H}_0|}{|\mathcal{S}_t| \vee 1}] \leq \delta$. Moreover, with probability at least $1 - 6\delta$, we have $\mathcal{H}_0 \cap \mathcal{R}_t = \emptyset$ for all $t \in \mathbb{N}$ and there exists a $T$ such that*

$$T \lesssim (n - |\mathcal{H}_1|)\Delta^{-2} \log(\max\{|\mathcal{H}_1|, \log\log(n/\delta)\} \log(\Delta^{-2})/\delta)$$
$$+ |\mathcal{H}_1|\Delta^{-2} \log(\max\{n - (1 - 2\delta(1 + 4\delta))|\mathcal{H}_1|, \log\log(n/\delta)\} \log(\Delta^{-2})/\delta)$$

*and $\mathcal{H}_1 \subseteq \mathcal{R}_t$ for all $t \geq T$. Note, together this implies $\mathcal{H}_1 = \mathcal{R}_t$ for all $t \geq T$.*

Theorem 4 has the strongest conditions, and therefore the largest sample complexity. Ignoring $\log\log\log(n)$ factors, $T$ roughly scales as $(n - |\mathcal{H}_1|) \log(|\mathcal{H}_1|) + |\mathcal{H}_1| \log(n - (1 - 2\delta(1 + 4\delta))|\mathcal{H}_1|)$. Inspecting the top-k lower bound of [8] where the arms' means in $\mathcal{H}_1$ are equal to $\mu_0 + \Delta$, the arms' means in $\mathcal{H}_0$ are equal to $\mu_0$, and the algorithm has knowledge of the cardinality of $\mathcal{H}_1$, a necessary sample complexity of $(n - |\mathcal{H}_1|) \log(|\mathcal{H}_1|) + |\mathcal{H}_1| \log(n - |\mathcal{H}_1|)$ is given. It is not clear whether this small difference of $\log(n - (1 - 2\delta(1 + 4\delta))|\mathcal{H}_1|)$ versus $\log(n - |\mathcal{H}_1|)$ is an artifact of our analysis, or a fundamental limitation when the cardinality $|\mathcal{H}_1|$ is unknown. We now state our final theorem.

**Theorem 5** (FWER, TPR). *For all $t$ we have $\mathbb{E}[\frac{|\mathcal{S}_t \cap \mathcal{H}_0|}{|\mathcal{S}_t| \vee 1}] \leq \delta$. Moreover, with probability at least $1 - 7\delta$ we have $\mathcal{H}_0 \cap \mathcal{R}_t = \emptyset$ for all $t \in \mathbb{N}$ and there exists a $T$ such that*

$$T \lesssim (n - |\mathcal{H}_1|)\Delta^{-2} \log(\log(\Delta^{-2})/\delta)$$
$$+ |\mathcal{H}_1|\Delta^{-2} \log(\max\{n - (1 - \eta)|\mathcal{H}_1|, \log\log(n \log(1/\delta)/\delta)\} \log(\Delta^{-2})/\delta)$$

*and $\mathbb{E}[\frac{|\mathcal{R}_t \cap \mathcal{H}_1|}{|\mathcal{H}_1|}] \geq 1 - \delta$ for all $t \geq T$, where $\eta = (1 - 3\delta - \sqrt{2\delta \log(1/\delta)/|\mathcal{H}_1|})$.*

# 4 Experiments

The distribution of each arm equals $\nu_i = \mathcal{N}(\mu_i, 1)$ where $\mu_i = \mu_0 = 0$ if $i \in \mathcal{H}_0$, and $\mu_i > 0$ if $i \in \mathcal{H}_1$. We consider three algorithms: $i$) uniform allocation with anytime BH selection as done in Algorithm 1, $ii$) successive elimination (SE) (see Appendix G)[5] that performs uniform allocation on only those arms that have not yet been selected by BH, and $iii$) Algorithm 1 (UCB). Algorithm 1 and the BH selection rule for all algorithms use $\phi(t, \delta) = \sqrt{\frac{2\log(1/\delta) + 6\log\log(1/\delta) + 3\log(\log(et/2))}{t}}$ from [25, Theorem 8]. In addition, we ran BH at level $\delta$ instead of $\delta/(6.4\log(36/\delta))$ as discussed in section 3. Here we present the sample complexity for TPR+FDR with $\delta = 0.05$ and different parameterizations of $\mu, n, |\mathcal{H}_1|$.

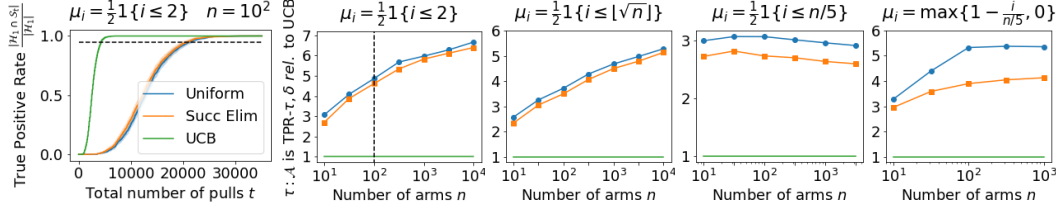

The first panel shows an empirical estimate of $\mathbb{E}\left[\frac{|\mathcal{S}_t \cap \mathcal{H}_1|}{|\mathcal{H}_1|}\right]$ at each time $t$ for each algorithm, averaged over 1000 trials. The black dashed line on the first panel denotes the level $\mathbb{E}\left[\frac{|\mathcal{S}_t \cap \mathcal{H}_1|}{|\mathcal{H}_1|}\right] = 1 - \delta = .95$, and corresponds to the dashed black line on the second panel. The right four panels show the number of samples each algorithm takes before the true positive rate exceeds $1 - \delta = .95$, relative to the number of samples taken by UCB, for various parameterizations. Panels two, three, and four have $\Delta_i = \Delta$ for $i \in \mathcal{H}_1$ while panel five is a case where the $\Delta_i$'s are linear for $i \in \mathcal{H}_1$. While the differences are most clear on the second panel when $|\mathcal{H}_1| = 2 = o(n)$, over all cases UCB uses at least $\approx 3$ times fewer samples than uniform and SE. For FDR+TPR, Appendix G shows uniform sampling roughly has a sample complexity that scales like $n\Delta^{-2}\log(\frac{n}{|\mathcal{H}_1|})$ while SE's is upper bounded by $\min\{n\Delta^{-2}\log(\frac{n}{|\mathcal{H}_1|}), (n - |\mathcal{H}_1|)\Delta^{-2}\log(\frac{n}{|\mathcal{H}_1|}) + \sum_{i \in \mathcal{H}_1}\Delta_i^{-2}\log(n)\}$. Comparing with Theorem 2 for the difference cases (i.e., $|\mathcal{H}_1| = 2, \sqrt{n}, n/5$) provides insight into the relative difference between UCB, uniform, and SE on the different panels.

## Acknowledgments

This work was informed and inspired by early discussions with Aaditya Ramdas on methods for controlling the false discovery rate (FDR) in multiple testing; we are grateful to have learned from a leader in the field. We also thank him for his careful reading and feedback. We'd also like to thank Martin J. Zhang for his input. We also thank the leading experimentation and A/B testing platform on the web, *Optimizely*, for its support, insight into its customers' needs, and for committing engineering time to implementing this research into their platform [12]. In particular, we thank Whelan Boyd, Jimmy Jin, Pete Koomen, Sammy Lee, Ajith Mascarenhas, Sonesh Surana, and Hao Xia at Optimizely for their efforts.

## Footnotes

[1]All results without modification apply to unbounded, sub-Gaussian random variables.

[2]All results generalize to the case when $\mathcal{H}_0 = \{i : \mu_i \leq \mu_0\}$.

[3] Each theorem relies on different events holding with high probability, and consequently a different $c$ for each. To have $c = 1$ for each of the four settings, we would have had to define different constants in the algorithm for each setting. We hope the reader forgives us for this attempt at minimizing clutter.

[4]In the asymptotic $n$ regime, it is common to study the case when $|\mathcal{H}_1| = n^\beta$ for $\beta \in (0, 1)$ [4, 13].

[5]Inspired by the best-arm identification literature [19].

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
