[Supplementary Material · bandit_FDR_finalJan11_full.pdf]

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

, see [21, 24, 25, 26]. Moreover, define its **inverse** $\phi^{-1}(\epsilon,\delta) = \min\{t : \phi(t,\delta) \leq \epsilon\}$. For the same constant $c_\phi$, it can be shown that $\phi^{-1}(\epsilon,\delta) \leq c_\phi \epsilon^{-2} \log(2\log(\frac{ec_\phi \epsilon^{-2}}{\delta})/\delta) \leq c\epsilon^{-2} \log(\log(\epsilon^{-2})/\delta)$ for a sufficiently large constant $c$ (and any $\epsilon, \delta < 1/4$).

The technical challenges in this work revolve around arguments that *avoid union bounds*. By union bounding over all $n$ arms we have

$$\mathbb{P}\left(\bigcup_{i=1}^{n}\bigcup_{t=1}^{\infty}\{|\widehat{\mu}_{i,t} - \mu_i| \leq \phi(t,\tfrac{\delta}{n})\}\right) \leq \sum_{i=1}^{n}\mathbb{P}\left(\bigcup_{t=1}^{\infty}\{|\widehat{\mu}_{i,t} - \mu_i| \leq \phi(t,\tfrac{\delta}{n})\}\right) \leq n\tfrac{\delta}{n} = \delta$$

which says that with probability at least $1 - \delta$, the deviations of *all* arms after any number of samples $s$ are bounded by $\phi(s, \delta/n)$. This is attractive because we can easily upper bound the number of times we need to sample an arm $i$ before its empirical mean is within $\epsilon$ of its true mean by $\phi^{-1}(\epsilon, \tfrac{\delta}{n}) \approx \epsilon^{-2} \log(\tfrac{n}{\delta} \log(\epsilon^{-2}))$. However, we note that this number of samples scales as $\log(n)$, and this $\log(n)$ will persist in any sample complexity result in any analysis that uses such a union bounding technique. That is, using union bounds in a naive way leads to sample complexities of at least $\Delta^{-2} n \log(n)$, which is no better than uniform sampling when considering FWER+FWPD (see Theorem 8)!

To avoid union bounds, we use techniques developed for best arm and top-k identification for multi-armed bandits [21, 8]. Define the random variable

$$\rho_i = \sup\left\{\rho \in (0,1] : \bigcap_{t=1}^{\infty}\{|\widehat{\mu}_{i,t} - \mu_i| \leq \phi(t,\rho)\}\right\} \tag{2}$$

and note that by the definition of $\phi$, we have $\mathbb{P}(\rho_i \leq x) \leq x$ for any $x \in (0,1]$. That is, each $\rho_i$ is an **independent sub-uniformly distributed random variable**. However, note that the $\rho_i$ random variables are *not* $p$-values, that is, they are unrelated to $P_{i,t}$. We will often make use of the fact that $\mu_i - \phi(t,\rho_i) \leq \widehat{\mu}_{i,t} \leq \mu_i + \phi(t,\rho_i)$ which is simply sandwiching a random quantity by two other random quantities. Furthermore, by definition $|\widehat{\mu}_{i,t} - \mu_i| \leq \phi(t,\delta)$ implies $\rho_i \geq \delta$. While a union bound would be equivalent to saying $\mathbb{P}(\bigcup_{i=1}^{n}\{\rho_i \leq \tfrac{\delta}{n}\}) \leq \delta$, we avoid union bounds by reasoning about only the statements we need. For instance, instead of guaranteeing *all* the $\rho_i$ are bounded by a single value, we will make guarantees about how *most* behave, and argue that this is sufficient. Examples of such statements are given in events $\mathcal{E}_3, \mathcal{E}_{4,0}, \mathcal{E}_{4,1}$ defined below. This strategy has been applied successfully to remove extraneous log factors (for instance, [8] improved the top-$k$ algorithm and analysis initially proposed in [9]).

Our proofs may upper bound an expression by another with a leading constant $c$, which may become $c'$ on the next line, then $c''$, and so on. This is meant to indicate that the constant is changing from line to line, but the next display may revert back to using $c$ and this is not the same $c$ as before. All constants are independent of problem parameters. This strategy is principally used to hide the ugliness of inverting the $\phi$ function, which arises only in the analysis and not the algorithm itself.

The proofs that follow are meant to be read sequentially, as some of the proofs will reuse the same lemmas. All four theorems claim that FDR is controlled for the set $\mathcal{S}_t$ in the algorithm. This is independent of the sampling scheme because of the defined anytime $p$-values.

# B  Anytime FDR for Anytime $p$-values

We begin our analysis by proving a few general statements about anytime $p$-values and FDR control for the set $\mathcal{S}_t$.

For p-values $p_1, \ldots, p_n$ and null set $\mathcal{H}_0 \subseteq [n]$ define

$$R_k = \{i : p_i \le \alpha \frac{k}{n}\}$$

$$FDP(R_k) = \frac{\sum_{i \in \mathcal{H}_0} 1\{p_i \le \alpha \frac{k}{n}\}}{|R_k| \vee 1}$$

$$\widehat{FDP}(R_k) = \frac{n \max_{i \in R_k} p_i}{|R_k| \vee 1}$$

The celebrated result of Benjamini-Hochberg says that under the assumption that the null p-values are independent and sub-uniformly distributed, if $\widehat{k} = \max\{k : \widehat{FDP}(R_k) \le \alpha\}$ then $\mathbb{E}[FDP(R_{\widehat{k}})] \le \alpha$. The following theorem provides a bound on the expected false discovery proportion for *any $k$* such that $\widehat{FDP}(R_k) \le \alpha$.

**Theorem 1.** *Fix $\alpha \in (0, e^{-1})$. For $n \ge 2$ let $p_1, \ldots, p_n$ be random variables such that $\{p_i\}_{i \in \mathcal{H}_0}$ are independent and sub-uniformly distributed so that $\max_{i \in \mathcal{H}_0} \mathbb{P}(p_i \le x) \le x$. For any $k \in \{0, 1, \ldots, n\}$, if $R_k = \{i : p_i \le \alpha \frac{k}{n}\}$ then*

$$\mathbb{E}\left[\max_{k : \widehat{FDP}(R_k) \le \alpha} FDP(R_k)\right] \le \frac{|\mathcal{H}_0|\alpha}{n} \left(2\log(\frac{2n}{|\mathcal{H}_0|\alpha}) + \log(8e^5 \log(\frac{8n}{|\mathcal{H}_0|\alpha}))\right)$$

$$\le 4\alpha \log(9/\alpha).$$

*In other words, any procedure that chooses a set $\{i : p_i \le \frac{\alpha k}{n}\}$ for any $k$ satisfying $|\{i : p_i \le \frac{\alpha k}{n}\}| \ge k$ is FDR controlled at level $O(\alpha \log(1/\alpha))$.*

*Proof.* Define

$$p_\ell^0 = \min\{t : \sum_{i \in \mathcal{H}_0} 1\{p_i \le t\} \ge \ell\}.$$

so that $p_\ell^0$ is the value of the $\ell$th largest p-value in $\mathcal{H}_0$. Note that

$$\max_{k : \widehat{FDP}(R_k) \le \alpha} FDP(R_k) = \max_{k : \widehat{FDP}(R_k) \le \alpha} \frac{\sum_{i \in \mathcal{H}_0} 1\{p_i \le \alpha \frac{k}{n}\}}{|R_k|}$$

$$= \max_{k : \widehat{FDP}(R_k) \le \alpha} \sum_{\ell=1}^{|\mathcal{H}_0|} \frac{\ell}{|R_k|} \mathbf{1}\left\{\ell = \sum_{i \in \mathcal{H}_0} 1\{p_i \le \alpha \frac{k}{n}\}\right\}$$

$$\le \max_{k : \widehat{FDP}(R_k) \le \alpha} \sum_{\ell=1}^{\ell_0} \frac{\ell}{|R_k|} \mathbf{1}\left\{\ell = \sum_{i \in \mathcal{H}_0} 1\{p_i \le \alpha \frac{k}{n}\}\right\}$$

$$+ \max_{k : \widehat{FDP}(R_k) \le \alpha} \sum_{\ell=\ell_0+1}^{|\mathcal{H}_0|} \frac{\ell}{|R_k|} \mathbf{1}\left\{\ell = \sum_{i \in \mathcal{H}_0} 1\{p_i \le \alpha \frac{k}{n}\}\right\}$$

for some $\ell_0 \in \{1, \ldots, |\mathcal{H}_0|\}$ to be defined later, where the last line follows from the fact that $\max_k (f(k) + g(k)) \le \max_k f(k) + \max_k g(k)$. If $\ell = \sum_{i \in \mathcal{H}_0} 1\{p_i \le \alpha \frac{k}{n}\}$ then $\max_{i \in R_k} p_i$ is at least as large as the $\ell$th largest p-value in $\{p_i : i \in \mathcal{H}_0\}$ (i.e., $p_\ell^0$) and

$$\frac{1\{\ell = \sum_{i \in \mathcal{H}_0} 1\{p_i \le \alpha \frac{k}{n}\}\}}{\max_{i \in R_k} p_i} \le \frac{1\{\ell = \sum_{i \in \mathcal{H}_0} 1\{p_i \le \alpha \frac{k}{n}\}\}}{p_\ell^0}.$$

Let $u_1, \ldots, u_{|\mathcal{H}_0|}$ be iid uniformly distributed random variables on $[0,1]$. Note that for every $i \in \mathcal{H}_0$ and $j = 1, \ldots, |\mathcal{H}_0|$ we have $\mathbb{P}(p_i \le x) \le x = \mathbb{P}(u_j \le x)$ where we have used the assumption that each $p_i$ is sub-uniform randomly distributed. Let $u_{(i)}$ be the $i$th largest $u_i$ such that $u_{(1)} \le u_{(2)} \le \cdots \le u_{(|\mathcal{H}_0|)}$. Note that for any $\ell = 1, \ldots, |\mathcal{H}_0|$ we have

$$\mathbb{P}\left(p_\ell^0 \le x\right) = \mathbb{P}(\sum_{i \in \mathcal{H}_0} 1\{p_i \le x\} \ge \ell) \le \mathbb{P}(\sum_{i=1}^{|\mathcal{H}_0|} \mathbf{1}\{u_i \le x\} \ge \ell) = \mathbb{P}\left(u_{(\ell)} \le x\right)$$

since each $\mathbf{1}\{p_i \leq x\}$ is an independent Bernoulli random variable with parameter $\mathbb{P}(p_i \leq x) \leq x = \mathbb{P}(u_j \leq x)$ for any $j \in \{1, \ldots, |\mathcal{H}_0|\}$.

We then observe that

$$\mathbb{P}\left(u_{(\ell)} \leq x\right) \leq \mathbb{P}(\sum_{i=1}^{|\mathcal{H}_0|} \mathbf{1}\{u_i \leq x\} \geq \ell)$$

$$\leq \exp\left(-|\mathcal{H}_0|KL(\tfrac{\ell}{|\mathcal{H}_0|}, x)\right)$$

$$\leq \exp\left(-|\mathcal{H}_0|\frac{\left(\frac{\ell}{|\mathcal{H}_0|} - x\right)^2}{2\ell/|\mathcal{H}_0|}\right)$$

$$= \exp\left(-\frac{\ell}{2}\left(1 - \frac{|\mathcal{H}_0|x}{\ell}\right)^2\right)$$

where the second inequality is Chernoff's bound, and the next line follows from $KL(x, y) \geq \frac{(x-y)^2}{2x}$ for $x > y$. For each $\ell = 1, 2, \ldots, |\mathcal{H}_0|$ pick $x = \frac{\ell}{2|\mathcal{H}_0|}$ so that $\mathbb{P}\left(p_\ell^0 \leq \frac{\ell}{2|\mathcal{H}_0|}\right) \leq e^{-\ell/8}$. Also, if we define the event

$$\mathcal{E}_\ell := \left\{p_\ell^0 \geq \frac{\ell}{2|\mathcal{H}_0|}\right\}$$

then $\mathbb{P}(\mathcal{E}_\ell) \geq 1 - e^{-\ell/8}$. If $\mathcal{E} = \cap_{\ell=\ell_0+1}^{|\mathcal{H}_0|} \mathcal{E}_\ell$ then

$$\max_{k:\widehat{FDP}(R_k) \leq \alpha} \sum_{\ell=\ell_0+1}^{|\mathcal{H}_0|} \frac{\ell\mathbf{1}\left\{\ell = \sum_{i \in \mathcal{H}_0} 1\{p_i \leq \alpha\frac{k}{n}\}\right\}}{|R_k|}$$

$$\leq \mathbf{1}\{\mathcal{E}^c\} + \mathbf{1}\{\mathcal{E}\} \max_{k:\widehat{FDP}(R_k) \leq \alpha} \sum_{\ell=\ell_0+1}^{|\mathcal{H}_0|} \frac{\ell\mathbf{1}\left\{\ell = \sum_{i \in \mathcal{H}_0} 1\{p_i \leq \alpha\frac{k}{n}\}\right\}}{|R_k|}$$

$$\leq \mathbf{1}\{\mathcal{E}^c\} + \mathbf{1}\{\mathcal{E}\} \max_{k:\widehat{FDP}(R_k) \leq \alpha} \sum_{\ell=\ell_0+1}^{|\mathcal{H}_0|} \frac{\ell\mathbf{1}\left\{\ell = \sum_{i \in \mathcal{H}_0} 1\{p_i \leq \alpha\frac{k}{n}\}\right\}}{n \max_{i \in R_k} p_i}\alpha$$

$$\leq \mathbf{1}\{\mathcal{E}^c\} + \mathbf{1}\{\mathcal{E}\} \max_{k:\widehat{FDP}(R_k) \leq \alpha} \sum_{\ell=\ell_0+1}^{|\mathcal{H}_0|} \frac{\ell\mathbf{1}\left\{\ell = \sum_{i \in \mathcal{H}_0} 1\{p_i \leq \alpha\frac{k}{n}\}\right\}}{np_\ell^0}\alpha$$

$$\leq \mathbf{1}\{\mathcal{E}^c\} + \mathbf{1}\{\mathcal{E}\} \max_k \sum_{\ell=\ell_0+1}^{|\mathcal{H}_0|} \frac{2|\mathcal{H}_0|\mathbf{1}\left\{\ell = \sum_{i \in \mathcal{H}_0} 1\{p_i \leq \alpha\frac{k}{n}\}\right\}}{n}\alpha$$

$$\leq \mathbf{1}\{\mathcal{E}^c\} + \mathbf{1}\{\mathcal{E}\}\frac{2|\mathcal{H}_0|}{n}\alpha \max_k \sum_{\ell=\ell_0+1}^{|\mathcal{H}_0|} \mathbf{1}\left\{\ell = \sum_{i \in \mathcal{H}_0} 1\{p_i \leq \alpha\frac{k}{n}\}\right\}$$

$$\leq \mathbf{1}\{\mathcal{E}^c\} + \frac{2|\mathcal{H}_0|}{n}\alpha.$$

Note that the only randomness on the right-hand-side is $\mathbf{1}\{\mathcal{E}^c\}$. After taking expectations on both sides we observe that

$$\mathbb{E}[\mathbf{1}\{\mathcal{E}^c\}] = \mathbb{E}\left[\mathbf{1}\{\cup_{\ell=\ell_0+1}^{|\mathcal{H}_0|}\mathcal{E}_\ell^c\}\right] = \sum_{\ell=\ell_0+1}^{|\mathcal{H}_0|} \mathbb{P}(\mathcal{E}_\ell^c) \leq \sum_{\ell=\ell_0+1}^{|\mathcal{H}_0|} e^{-\ell/8} \leq 8e^{-\ell_0/8}.$$

Thus,

$$\mathbb{E}\left[\max_{k:\widehat{FDP}(R_k) \leq \alpha} \sum_{\ell=\ell_0+1}^{|\mathcal{H}_0|} \frac{\ell\mathbf{1}\left\{\ell = \sum_{i \in \mathcal{H}_0} 1\{p_i \leq \alpha\frac{k}{n}\}\right\}}{|R_k|}\right] \leq 8e^{-\ell_0/8} + \frac{2|\mathcal{H}_0|}{n}\alpha.$$

On the other hand, for any $k$ with $\widehat{FDP}(R_k) = \frac{n\max_{i \in R_k} p_i}{|R_k| \vee 1} \leq \alpha$ and $\ell = \sum_{i \in \mathcal{H}_0} 1\{p_i \leq \alpha\frac{k}{n}\}$ we have that the $\ell$th $p$-value is in $R_k$ and $\frac{np_\ell^0}{\alpha} \leq \frac{n\max_{i \in R_k} p_i}{\alpha} \leq |R_k|$ where $|R_k| \geq 1$ necessarily. Consequently, $|R_k| \geq \lceil np_\ell^0/\alpha \rceil$ so that

$$\max_{k:\widehat{FDP}(R_k)\leq\alpha} \sum_{\ell=1}^{\ell_0} \frac{\ell}{|R_k|} \mathbf{1}\left\{\ell = \sum_{i \in \mathcal{H}_0} 1\{p_i \leq \alpha\tfrac{k}{n}\}\right\} \leq \max_{k:\widehat{FDP}(R_k)\leq\alpha} \sum_{\ell=1}^{\ell_0} \frac{\ell}{|R_k|} \mathbf{1}\left\{\ell = \sum_{i \in \mathcal{H}_0} 1\{p_i \leq \alpha\tfrac{k}{n}\}\right\}$$

$$\leq \sum_{\ell=1}^{\ell_0} \max_k \frac{\ell}{|R_k|} \mathbf{1}\left\{\ell = \sum_{i \in \mathcal{H}_0} 1\{p_i \leq \alpha\tfrac{k}{n}\}\right\}$$

$$\leq \sum_{\ell=1}^{\ell_0} \max_k \frac{\ell}{\lceil np_\ell^0/\alpha \rceil} \mathbf{1}\left\{\ell = \sum_{i \in \mathcal{H}_0} 1\{p_i \leq \max_{i \in R_k} p_i\}\right\}$$

$$= \sum_{\ell=1}^{\ell_0} \frac{\ell}{\lceil np_\ell^0/\alpha \rceil}$$

As above, we use the fact that each $p_\ell^0$ is stochastically dominated by $u_{(\ell)}$ so that

$$\mathbb{E}\left[\frac{\ell}{\lceil np_\ell^0/\alpha \rceil}\right] = \int_{x=0}^{\infty} \mathbb{P}\left(\frac{\ell}{\lceil np_\ell^0/\alpha \rceil} \geq x\right) dx$$

$$= \int_{x=0}^{\infty} \mathbb{P}\left(\lceil np_\ell^0/\alpha \rceil \leq \ell/x\right) dx$$

$$= \int_{x=0}^{\infty} \mathbb{P}\left(np_\ell^0/\alpha \leq \lfloor \ell/x \rfloor\right) dx$$

$$\leq \int_{x=0}^{\infty} \mathbb{P}\left(nu_{(\ell)}/\alpha \leq \lfloor \ell/x \rfloor\right) dx$$

$$= \int_{x=0}^{\infty} \mathbb{P}\left(\lceil nu_{(\ell)}/\alpha \rceil \leq \ell/x\right) dx$$

$$= \mathbb{E}\left[\frac{\ell}{\lceil nu_{(\ell)}/\alpha \rceil}\right]$$

since $\lceil a \rceil > b \iff a > \lfloor b \rfloor$. Thus,

$$\mathbb{E}\left[\max_{k:\widehat{FDP}(R_k)\leq\alpha} \sum_{\ell=1}^{\ell_0} \frac{\ell}{|R_k|} \mathbf{1}\left\{\ell = \sum_{i \in \mathcal{H}_0} 1\{p_i \leq \alpha\tfrac{k}{n}\}\right\}\right] \leq \sum_{\ell=1}^{\ell_0} \mathbb{E}\left[\frac{\ell}{\lceil np_\ell^0/\alpha \rceil}\right] \leq \sum_{\ell=1}^{\ell_0} \mathbb{E}\left[\frac{\ell}{\lceil nu_{(\ell)}/\alpha \rceil}\right].$$

For notational ease, let $m = |\mathcal{H}_0|$. Recall that the PDF of the $\ell$-th order statistic of $m$ iid uniform random variables on $[0,1]$ is given by,

$$\frac{d\mathbb{P}(u_{(\ell)} \leq x)}{dx} = \ell\binom{m}{\ell}(1-x)^{m-\ell}x^{\ell-1}.$$

The following simple bound will be useful for small $\ell$ and $y$:

$$\mathbb{P}(u_{(\ell)} \leq y) = \int_{x=0}^{y} \ell\binom{m}{\ell}(1-x)^{m-\ell}x^{\ell-1}dx \leq \binom{m}{\ell}y^\ell \leq (my)^\ell.$$

First we consider the case when $\ell > 1$.

$$\mathbb{E}\left[\frac{\ell}{\lceil nu_{(\ell)}/\alpha\rceil}\right] = \sum_{k=1}^{\infty}\frac{1}{k}\mathbb{P}\left(\frac{(k-1)\alpha}{n} \le u_{(\ell)} \le \frac{k\alpha}{n}\right)$$

$$\le \mathbb{P}\left(u_{(\ell)} \le \frac{\alpha}{n}\right) + \int_{x=\frac{\alpha}{n}}^{1}\frac{\alpha}{nx}d\mathbb{P}(u_{(\ell)} \le x)$$

$$= \mathbb{P}\left(u_{(\ell)} \le \frac{\alpha}{n}\right) + \frac{\alpha}{n}\ell\binom{m}{\ell}\int_{x=\frac{\alpha}{n}}^{1}(1-x)^{m-\ell}x^{\ell-2}dx$$

$$\le \left(\frac{\alpha m}{n}\right)^{\ell} + \frac{\alpha}{n}\frac{\ell\binom{m}{\ell}}{(\ell-1)\binom{m-1}{\ell-1}}\int_{\alpha/n}^{1}(\ell-1)\binom{m-1}{\ell-1}(1-x)^{m-\ell}x^{\ell-2}dx$$

$$\le \left(\frac{\alpha m}{n}\right)^{\ell} + \frac{\alpha}{n}\frac{m!}{(\ell-1)!(m-\ell)!}\frac{(\ell-2)!(m-\ell)!}{(m-1)!}$$

$$= \left(\frac{\alpha m}{n}\right)^{\ell} + \frac{\alpha}{n}\frac{m}{\ell-1}$$

where going from the third line to the fourth, the integrand is just the PDF of a Beta-distribution so the integral can be bounded away by 1. When $\ell = 1$, the argument is slightly different. Firstly note that the PDF is a decreasing function, hence

$$\mathbb{P}\left(\frac{(k-1)\alpha}{n} \le u_{(1)} \le \frac{k\alpha}{n}\right) \le \frac{\alpha}{n}m\left(1 - \frac{\alpha(k-1)}{n}\right)^{m-1}$$

$$\le \frac{\alpha}{n}me^{-(m-1)\frac{\alpha(k-1)}{n}}$$

using the fact that $e^{-x} > 1 - x$. Hence,

$$\mathbb{E}\left[\frac{1}{\lceil nu_{(1)}/\alpha\rceil}\right] = \sum_{k=1}^{\lceil n/\alpha\rceil}\frac{1}{k}\mathbb{P}\left(\frac{(k-1)\alpha}{n} \le u_{(1)} \le \frac{k\alpha}{n}\right)$$

$$\le \frac{m\alpha}{n}\sum_{k=1}^{\infty}\frac{1}{k}e^{-(m-1)\frac{\alpha(k-1)}{n}}$$

$$= \frac{m\alpha}{n}(1 + \sum_{k=1}^{\infty}\frac{1}{k+1}e^{-\frac{(m-1)\alpha}{n}k})$$

$$= \frac{m\alpha}{n}e^{\frac{(m-1)\alpha}{n}}\log\left(\frac{1}{1-e^{-\frac{(m-1)\alpha}{n}}}\right)$$

$$\le 2\frac{m\alpha}{n}\log(\frac{n}{(m-1)\alpha})$$

where the last line holds for $\alpha(m-1)/n \le e^{-1}$ by observing that $e^x\log(1-e^{-x})/\log(x)$ is increasing and less than 2 at $x = e^{-1}$. On the other hand, if $m = 1$ then in the above display $\sum_{k=1}^{\lceil n/\alpha\rceil}1/k \le \log(\lceil n/\alpha\rceil e) \le 2\log(n/\alpha)$ for $\alpha \le e^{-1}$ and $n \ge 2$. Thus,

$$\mathbb{E}\left[\frac{1}{\lceil nu_{(1)}/\alpha\rceil}\right] = \sum_{k=1}^{\lceil n/\alpha\rceil}\frac{1}{k}\mathbb{P}\left(\frac{(k-1)\alpha}{n} \le u_{(1)} \le \frac{k\alpha}{n}\right)$$

$$\le \begin{cases} 2\frac{m\alpha}{n}\log(\frac{n}{(m-1)\alpha}) & \text{if } m > 1 \\ 2\frac{\alpha}{n}\log(\frac{n}{\alpha}) & \text{if } m = 1 \end{cases}$$

$$\le 2\frac{m\alpha}{n}\log(\frac{2n}{m\alpha})$$

Plugging it all in we get

$$\sum_{\ell=1}^{\ell_0}\mathbb{E}\left[\frac{\ell}{\lceil nu_{(\ell)}/\alpha\rceil}\right] \le 2\frac{m\alpha}{n}\log(\frac{2n}{m\alpha}) + \sum_{\ell=2}^{\ell_0}\left(\left(\frac{\alpha m}{n}\right)^{\ell} + \frac{\alpha}{n}\frac{m}{\ell-1}\right)$$

$$\le 2\frac{m\alpha}{n}\log(\frac{2n}{m\alpha}) + \left(\frac{\alpha m}{n}\right)^{2}/(1-\frac{\alpha m}{n}) + \frac{\alpha m}{n}\log(\ell_0 e)$$

Putting it all together, we conclude that

$$\mathbb{E}\left[\max_{k:\widehat{FDP}(R_k)\leq\alpha} FDP(R_k)\right] \leq 8e^{-\ell_0/8} + \frac{2|\mathcal{H}_0|}{n}\alpha + \sum_{\ell=1}^{\ell_0}\mathbb{E}\left[\frac{\ell}{\lceil nu_{(\ell)}/\alpha\rceil}\right]$$

$$\leq 8e^{-\ell_0/8} + \frac{|\mathcal{H}_0|}{n}\alpha\left(2\log(\tfrac{2n}{|\mathcal{H}_0|\alpha}) + \log(\ell_0 e^4)\right)$$

$$\leq \frac{|\mathcal{H}_0|\alpha}{n}\left(2\log(\tfrac{2n}{|\mathcal{H}_0|\alpha}) + \log(8e^5\log(\tfrac{8n}{|\mathcal{H}_0|\alpha}))\right)$$

$$= O\left(\frac{|\mathcal{H}_0|\alpha}{n}\log(\tfrac{n}{|\mathcal{H}_0|\alpha})\right)$$

Where the last lines have taken $\ell_0 = 8\log(\frac{8n}{|\mathcal{H}_0|\alpha})$. $\qquad\square$

In this section, consider a hypothesis test between $\mathcal{H}_0$ and $\mathcal{H}_1$. Let $\{P_{i,t}\}_{t=1}^{\infty}, P_{i,t} \in (0,1]$ be a collection of random variables such that for $i \in \mathcal{H}_0$, $\{P_{i,t}\}_{i=1}^{\infty}$ are sub-uniformly distributed anytime $p$-values, i.e. $\mathbb{P}(\cup_{t=1}^{\infty}\{P_{i,t} \leq x\}) \leq x$. Let $\mathcal{S}_t$ be the set of discoveries following the Benjamini-Hochberg procedure at confidence level $\delta$ at time $t$ on the $p$-values $P_{i,T_i(t)}, 1 \leq i \leq n$. The following lemma employees the previous Theorem to guarantees FDR-control.

**Lemma 1.** *For all times $t \geq 1$, FDR is controlled at level $\delta$ so that $\mathbb{E}[\frac{|\mathcal{S}_t \cap \mathcal{H}_0|}{|\mathcal{S}_t|\vee 1}] \leq \delta$.*

*Proof.* For any $i \in \mathcal{H}_0$ so that $\mu_i \leq \mu_0$ define $P_{i,*} = \inf_{t\geq 1} P_{i,t}$. Then for any $i \in \mathcal{H}_0$ and $\delta \in (0,1)$

$$\mathbb{P}(P_{i,*} \leq \delta) = \mathbb{P}\left(\bigcup_{t=1}^{\infty}\{P_{i,t} \leq \delta\}\right) = \mathbb{P}\left(\bigcup_{t=1}^{\infty}\{\widehat{\mu}_{i,t} - \phi(t,\delta) \geq \mu_0\}\right)$$

$$\leq \mathbb{P}\left(\bigcup_{t=1}^{\infty}\{\widehat{\mu}_{i,t} - \phi(t,\delta) \geq \mu_i\}\right) \leq \mathbb{P}\left(\bigcup_{t=1}^{\infty}\{|\widehat{\mu}_{i,t} - \mu_i| \geq \phi(t,\delta)\right) \leq \delta.$$

We observe that $\{P_{i,*}\}_{i\in\mathcal{H}_0}$ are sub-uniformly distributed random variables but are also *independent* since they only depend on the rewards of arm $i \in \mathcal{H}_0$. Thus, we may apply the above proposition at time $t$ with $\{P_{i,*}: i \in \mathcal{H}_0\} \cup \{P_{i,T_i(t)}: i \in \mathcal{H}_1\}$ to conclude that FDR would be controlled for any of the prescribed values of $k$ (including the largest, which would be equivalent to BH). We need to show that the BH procedure controls FDR at time $t$ with $\{P_{i,T_i(t)}: i \in \mathcal{H}_0\} \cup \{P_{i,T_i(t)}: i \in \mathcal{H}_1\}$.

As described in Section 2.1, the BH procedure selects the $\widehat{k}$ smallest $p$-values where

$$\widehat{k} = \max\{k: |\{i: P_{i,T_i(t)} \leq \delta\tfrac{k}{n}\}| \geq k\}.$$

By definition

$$\widehat{k} = |\{i \in \mathcal{H}_0: P_{i,T_i(t)} \leq \delta\tfrac{\widehat{k}}{n}\} \cup \{i \in \mathcal{H}_1: P_{i,T_i(t)} \leq \delta\tfrac{\widehat{k}}{n}\}|$$

$$\leq |\{i \in \mathcal{H}_0: P_{i,*} \leq \delta\tfrac{\widehat{k}}{n}\} \cup \{i \in \mathcal{H}_1: P_{i,T_i(t)} \leq \delta\tfrac{\widehat{k}}{n}\}|$$

since $P_{i,*} \leq P_{i,T_i(t)}$ for all $i \in \mathcal{H}_0$. Thus, since the $\{P_{i,*}\}_{i\in\mathcal{H}_0}$ are independent and sub-uniformly distributed and *at least* $\widehat{k}$ of $\{i \in \mathcal{H}_0: P_{i,*} \leq \delta\tfrac{\widehat{k}}{n}\} \cup \{i \in \mathcal{H}_1: P_{i,T_i(t)} \leq \delta\tfrac{\widehat{k}}{n}\}$ are below the threshold $\frac{\widehat{k}\delta}{n}$ we apply Theorem1 to conclude that the FDR of $\{i \in \mathcal{H}_0: P_{i,*} \leq \delta\tfrac{\widehat{k}}{n}\} \cup \{i \in \mathcal{H}_1: P_{i,T_i(t)} \leq \delta\tfrac{\widehat{k}}{n}\}$ is bounded by $\delta$.

We observe that

$$FDR(\{i \in [n]: P_{i,T_i(t)} \leq \delta\tfrac{\widehat{k}}{n}\}) = \mathbb{E}\left[\frac{\sum_{i\in\mathcal{H}_0}\mathbf{1}\{P_{i,T_i(t)} \leq \delta\tfrac{\widehat{k}}{n}\}}{\sum_{i\in\mathcal{H}_0}\mathbf{1}\{P_{i,T_i(t)} \leq \delta\tfrac{\widehat{k}}{n}\} + \sum_{i\in\mathcal{H}_1}\mathbf{1}\{P_{i,T_i(t)} \leq \delta\tfrac{\widehat{k}}{n}\}}\right]$$

$$\leq \mathbb{E}\left[\frac{\sum_{i\in\mathcal{H}_0}\mathbf{1}\{P_{i,*} \leq \delta\tfrac{\widehat{k}}{n}\}}{\sum_{i\in\mathcal{H}_0}\mathbf{1}\{P_{i,*} \leq \delta\tfrac{\widehat{k}}{n}\} + \sum_{i\in\mathcal{H}_1}\mathbf{1}\{P_{i,T_i(t)} \leq \delta\tfrac{\widehat{k}}{n}\}}\right]$$

$$= FDR(\{i \in \mathcal{H}_0: P_{i,*} \leq \delta\tfrac{\widehat{k}}{n}\} \cup \{i \in \mathcal{H}_1: P_{i,T_i(t)} \leq \delta\tfrac{\widehat{k}}{n}\})$$

$$\leq \delta$$

which is precisely what we wished to prove, where the first inequality follows from $\frac{a}{a+c} \leq \frac{a+b}{a+b+c}$ for positive numbers $a, b, c$, and the fact that $\sum_{i \in \mathcal{H}_0} \mathbf{1}\{P_{i,T_i(t)} \leq \delta \frac{\widehat{k}}{n}\} \leq \sum_{i \in \mathcal{H}_0} \mathbf{1}\{P_{i,*} \leq \delta \frac{\widehat{k}}{n}\}$ due to $P_{i,*} \leq P_{i,T_i(t)}$ for all $i \in \mathcal{H}_0$.

□

The following lemma is a stronger result, providing an anytime high-probability bound on the false discovery proportion and the size of the discovered set. The proof follows from considering not the $p$-values, $P_{i,t}$ at any given time, but rather the random variable which is the worst-case $p$-values over all time.

**Lemma 2.** *Let $T_i : \mathbb{N} \to \mathbb{N}$ be an arbitrary function for any $i$. Recall $\delta' = \delta/(6.4 \log(36/\delta))$. Then with probability greater than $\geq 1 - \delta'$,*

$$\bigcap_{t=1}^{\infty} \left\{ |\mathcal{S}_t| \leq \frac{1}{1 - 2\delta'(1+4\delta')} |\mathcal{H}_1| + \frac{4(1+4\delta')/3}{1 - 2\delta'(1+4\delta')} \log\left(\frac{5 \log_2(n/\delta')}{\delta'}\right) \right\}$$

*and*

$$\bigcap_{t=1}^{\infty} \left\{ \frac{|\mathcal{S}_t \cap \mathcal{H}_0|}{|\mathcal{S}_t| \vee 1} \leq \delta' \frac{|\mathcal{H}_0|}{n} + (1 + 4\delta') \sqrt{\frac{4\delta' \frac{|\mathcal{H}_0|}{n} \log\left(\frac{\log_2(n/\delta)}{\delta'}\right)}{|\mathcal{S}_t| \vee 1}} + \frac{(1 + 4\delta') \log\left(\frac{\log_2(n/\delta')}{\delta'}\right)}{3(|\mathcal{S}_t| \vee 1)} \right\}$$

*in particular, these events hold on $\mathcal{E}_3$ (defined in the proof), which holds with probability greater than $1 - \delta'$.*

*Proof.* As in the proof of Lemma 1, let $P_{i,*} = \inf_{t \geq 1} P_{i,t}$ for all $i \in \mathcal{H}_0$.

Define the event,

$$\mathcal{E}_3 := \left\{ \forall s \in (0, 1] : \sum_{i \in \mathcal{H}_0} \mathbf{1}\{P_{i,*} \leq s\} \right.$$

$$\left. \leq s|\mathcal{H}_0| + (1 + 4s)\sqrt{2 \max\{2s, 2\delta'/n\}|\mathcal{H}_0| \log\left(\frac{\log_2(n/\delta')}{\delta'}\right)} + \frac{1+4s}{3} \log\left(\frac{\log_2(n/\delta')}{\delta'}\right) \right\}$$

We have that $\mathbb{P}(\mathcal{E}_3) \geq 1 - \delta'$ by applying Lemma 9 found in the appendix with $c = 2\delta'/n$ and $X_i = P_{i,*}$ for $i \in \mathcal{H}_0$ so that $m = |\mathcal{H}_0|$. Note that there exists some threshold $\tau_t \in [0, 1]$ such that $BH$ selects all indices with $P_{i,T_i(t)} \leq \tau_t$ so that $\mathcal{S}_t = \mathcal{S}(\tau_t) := \{i \in [n] : P_{i,T_i(t)} \leq \tau_t\}$. Then

$$|\mathcal{S}_t \cap \mathcal{H}_0| = \sum_{i \in \mathcal{H}_0} \mathbf{1}\{P_{i,T_i(t)} \leq \tau_t\} \leq \sum_{i \in \mathcal{H}_0} \mathbf{1}\{P_{i,*} \leq \tau_t\}.$$

By definition, $\tau_t = \sup\{s \leq 1 : \frac{|\mathcal{S}(s)|}{n} \delta' \geq s\}$, otherwise we take it to be 0, so that $\tau_t \leq \delta'|\mathcal{S}(\tau_t)|/n$. We apply this inequality and $\mathcal{E}_3$ to observe

$$|\mathcal{S}_t \cap \mathcal{H}_0| \leq \tau_t|\mathcal{H}_0| + (1 + 4\tau_t)\sqrt{2 \max\{2\delta'/n, 2\tau_t\}|\mathcal{H}_0| \log\left(\frac{\log_2(n/\delta')}{\delta'}\right)} + \frac{1+4\tau_t}{3} \log\left(\frac{\log_2(n/\delta')}{\delta'}\right)$$

$$\leq \delta'|\mathcal{S}(\tau_t)|\frac{|\mathcal{H}_0|}{n} + (1 + 4\delta')\sqrt{2 \max\{2\delta'/n, 2\delta'|\mathcal{S}(\tau_t)|\frac{1}{n}\}|\mathcal{H}_0| \log\left(\frac{\log_2(n/\delta')}{\delta'}\right)} + \frac{1+4\delta'}{3} \log\left(\frac{\log_2(n/\delta')}{\delta'}\right)$$

$$= \delta'|\mathcal{S}(\tau_t)|\frac{|\mathcal{H}_0|}{n} + 2(1 + 4\delta')\sqrt{\delta'|\mathcal{S}(\tau_t)|\frac{|\mathcal{H}_0|}{n} \log\left(\frac{\log_2(n/\delta')}{\delta'}\right)} + \frac{1+4\delta'}{3} \log\left(\frac{\log_2(n/\delta')}{\delta'}\right)$$

$$\leq 2\delta'(1 + 4\delta')|\mathcal{S}(\tau_t)|\frac{|\mathcal{H}_0|}{n} + \frac{4(1+4\delta')}{3} \log\left(\frac{\log_2(n)}{\delta'}\right)$$

where the last inequality follows from $a + 2\sqrt{ab} + b = (\sqrt{a} + \sqrt{b})^2 \leq 2(a + b)$. After rearranging, because $|\mathcal{S}_t| = |\mathcal{S}_t \cap \mathcal{H}_1| + |\mathcal{S}_t \cap \mathcal{H}_0| \leq |\mathcal{H}_1| + |\mathcal{S}_t \cap \mathcal{H}_0|$ we obtain the theorem. □

We will refer to event $\mathcal{E}_3$ defined in the proof above in what follows.

# C  Proof of Theorem 2

We will need the following event. Let $a \in \mathbb{R}_+^n$ be a fixed vector to be defined later in the proof and define

$$\mathcal{E}_{4,j} := \left\{ \sum_{i \in \mathcal{H}_j} a_i \log(1/\rho_i) \leq 5 \log(1/\delta) \sum_{i \in \mathcal{H}_j} a_i \right\} \quad j \in \{0, 1\}$$

**Lemma 3.** $\min\{\mathbb{P}(\mathcal{E}_{4,0}), \mathbb{P}(\mathcal{E}_{4,1})\} \geq 1 - \delta.$

*Proof.* The proof is the same for $\mathcal{H}_0$ and $\mathcal{H}_1$ so we prove it for $i = 1, \ldots, n$. Because for $i = 1, \ldots, n$ the $\rho_i$ are independent, sub-uniformly distributed random variables, we have that $Z_i = a_i \log(1/\rho_i)$ are independent random variables satisfying $\mathbb{P}(Z_i \geq t) \leq \exp(-t/a_i)$. To see this, $\mathbb{P}(\rho_i \leq x) \leq x$ by definition, and thus $\mathbb{P}(a_i \log(1/\rho_i) \geq a_i \log(1/x)) \leq x$. Set $t = a_i \log(1/x)$ and solve for $x$. The result follows by applying Lemma 8. $\qquad\square$

We restate the theorem using the above events.

**Theorem 2** (FDR, TPR). *Let $\mathcal{H}_1 = \{i \in [n] : \mu_i > \mu_0\}$, $\mathcal{H}_0 = \{i \in [n] : \mu_i \leq \mu_0\}$. Define $\Delta_i = \mu_i - \mu_0$ for $i \in \mathcal{H}_1$, $\Delta_i = \min_{j \in \mathcal{H}_1} \mu_j - \mu_i$ for $i \in \mathcal{H}_0$, and $\Delta = \min_{i \in \mathcal{H}_1} \Delta_i$. For all $t$ we have $\mathbb{E}[\frac{|\mathcal{S}_t \cap \mathcal{H}_0|}{|\mathcal{S}_t|}] \leq \delta$. Moreover, on $\mathcal{E}_{4,0} \cap \mathcal{E}_{4,1}$ (which holds with probability at least $1 - 2\delta$), there exists a $T$ such that*

$$T \lesssim \sum_{i \in \mathcal{H}_0} \Delta_i^{-2} \log(\log(\Delta_i^{-2})/\delta) + \sum_{i \in \mathcal{H}_1} \Delta_i^{-2} \log(n \log(\Delta_i^{-2})/\delta)$$

*and $\mathbb{E}[\frac{|\mathcal{S}_t \cap \mathcal{H}_1|}{|\mathcal{H}_1|}] \geq 1 - \delta$ for all $t \geq T$. Also, on the same events there exists a $T$ such that*

$$T \lesssim n\Delta^{-2} \log(\log(\Delta^{-2})/\delta)$$

*and $\mathbb{E}[\frac{|\mathcal{S}_t \cap \mathcal{H}_1|}{|\mathcal{H}_1|}] \geq 1 - \delta$ for all $t \geq T$. Note, neither follows from the other.*

*Proof.* Define the random set $\mathcal{I} = \{i \in \mathcal{H}_1 : \rho_i \geq \delta\}$. Note, this is equivalent to $\mathcal{I} = \{i \in \mathcal{H}_1 : \widehat{\mu}_{i,T_i(t)} + \phi(T_i(t), \delta) \geq \mu_i \; \forall t \in \mathbb{N}\}$ since if $\rho_i \geq \delta$ then $\phi(T_i(t), \delta) > \phi(T_i(t), \rho_i)$ and

$$\widehat{\mu}_{i,T_i(t)} + \phi(T_i(t), \delta) \geq \mu_i + \phi(T_i(t), \delta) - \phi(T_i(t), \rho_i) \geq \mu_i.$$

Our aim is to show that this "well-behaved" set of indices $\mathcal{I}$ will be added to $\mathcal{S}_t$ in the claimed amount of time. This is sufficient for the TPR result because $\mathbb{E}|\mathcal{I}| = \sum_{i \in \mathcal{H}_1} \mathbb{E}[\mathbf{1}\{\rho_i > \delta\}] = \sum_{i \in \mathcal{H}_1} \mathbb{P}(\rho_i > \delta) \geq (1 - \delta)|\mathcal{H}_1|$ since each $\rho_i$ is a sub-uniformly distributed random variable. To be clear, we are not claiming which *particular* indices of $\mathcal{H}_1$ will be added to $\mathcal{S}_t$ (indeed, $\mathcal{I}$ is a random set), just that their number exceeds $(1 - \delta)|\mathcal{H}_1|$ in expectation.

Note that $\mathcal{S}_t \subseteq \mathcal{S}_{t+1}$ for all $t$ so define $T = \min\{t \in \mathbb{N} : \mathcal{I} \subseteq \mathcal{S}_{t+1}\}$ if such inclusion ever exists, otherwise let $T = \infty$. Then

$$T = \sum_{t=1}^{\infty} \mathbf{1}\{\mathcal{I} \not\subseteq \mathcal{S}_t\}$$

$$= \sum_{t=1}^{\infty} \mathbf{1}\{I_t \in \mathcal{H}_0, \mathcal{I} \not\subseteq \mathcal{S}_t\} + \mathbf{1}\{I_t \in \mathcal{H}_1\}$$

We will bound each sum separately, starting with the first.

For any $j \in \mathcal{I}$ we have

$$\widehat{\mu}_{j,T_j(t)} + \phi(T_j(t), \delta) \geq \mu_j + \phi(T_j(t), \delta) - \phi(T_j(t), \rho_j) \geq \mu_j.$$

Thus $\{\mathcal{I} \not\subseteq \mathcal{S}_t\}$ implies that

$$\arg\max_{j \in \mathcal{S}_t^c} \widehat{\mu}_{j,T_j(t)} + \phi(T_j(t), \delta) \geq \min_{j \in \mathcal{I}} \mu_j \geq \min_{j \in \mathcal{H}_1} \mu_j.$$

On the other hand, for any $i \in \mathcal{H}_0$ we have

$$\widehat{\mu}_{i,T_i(t)} + \phi(T_i(t), \delta) \leq \mu_i + \phi(T_i(t), \delta) + \phi(T_i(t), \rho_i)$$
$$\leq \mu_i + 2\phi(T_i(t), \delta\rho_i)$$

so that $\widehat{\mu}_{i,T_i(t)} + \phi(T_i(t), \delta) \leq \min_{j \in \mathcal{H}_1} \mu_j = \mu_i + \Delta_i$ whenever $T_i(t) \geq \phi^{-1}(\frac{\Delta_i}{2}, \delta\rho_i)$. If $T_i(t)$ were this large, the arm $i \in \mathcal{H}_0$ could not be pulled because its upper confidence bound would be below the upper confidence bound of an arm $j \in \mathcal{I} \cap \mathcal{S}_t^c$. Thus,

$$\sum_{t=1}^{\infty} \mathbf{1}\{I_t \in \mathcal{H}_0, \mathcal{I} \not\subseteq \mathcal{S}_t\} \leq \sum_{i \in \mathcal{H}_0} \phi^{-1}(\tfrac{\Delta_i}{2}, \delta\rho_i)$$
$$\leq \sum_{i \in \mathcal{H}_0} c\Delta_i^{-2} \log(\log(\Delta_i^{-2})/(\delta\rho_i))$$
$$\leq \sum_{i \in \mathcal{H}_0} c\Delta_i^{-2} \log(\log(\Delta_i^{-2})/\delta) + c\Delta_i^{-2} \log(1/\rho_i)$$
$$\overset{\mathcal{E}_{4,0}}{\leq} \sum_{i \in \mathcal{H}_0} c\Delta_i^{-2} \log(\log(\Delta_i^{-2})/\delta) + 5c\Delta_i^{-2} \log(1/\delta)$$
$$\leq \sum_{i \in \mathcal{H}_0} c'\Delta_i^{-2} \log(\log(\Delta_i^{-2})/\delta)$$
$$\leq c''|\mathcal{H}_0|\Delta^{-2} \log(\log(\Delta^{-2})/\delta)$$

which concludes the upper bound on the first term.

For the second term, consider a time that $I_t \in \mathcal{H}_1$. Recall $\delta' = \delta/(9.6 \log(3000/\delta))$. For any $j \in \mathcal{H}_1$ and arbitrary $k \leq n$ by the BH procedure in the algorithm we have

$$\widehat{\mu}_{j,T_j(t)} - \phi(T_j(t), \delta'\tfrac{k}{n}) \geq \mu_j - \phi(T_j(t), \delta'\tfrac{k}{n}) - \phi(T_j(t), \rho_j)$$
$$\geq \mu_j - 2\phi(T_j(t), \delta'\rho_j\tfrac{k}{n})$$

so that $\widehat{\mu}_{j,T_j(t)} - \phi(T_j(t), \delta'\rho_j\tfrac{k}{n}) \geq \mu_0$ whenever $T_j(t) \geq \phi^{-1}(\frac{\mu_j-\mu_0}{2}, \delta'\rho_j\frac{k}{n})$, guaranteeing its spot in $s(k)$. In the worst case, the arms are added one at a time to $s(k)$, instead as a group. Thus, if $\pi$ is any map $\mathcal{H}_1 \rightarrow \{1, \ldots, |\mathcal{H}_1|\}$ then

$$\sum_{t=1}^{\infty} \mathbf{1}\{I_t \in \mathcal{H}_1\} \leq \max_{\pi} \sum_{i \in \mathcal{H}_1} \phi^{-1}(\tfrac{\mu_i-\mu_0}{2}, \delta'\rho_i\tfrac{\pi(i)}{n})$$
$$\leq \max_{\pi} \sum_{i \in \mathcal{H}_1} \left(c\Delta_i^{-2} \log(\tfrac{n}{\pi(i)} \log(\Delta_i^{-2})/\delta') + c\Delta_i^{-2} \log(1/\rho_i)\right)$$
$$\overset{\mathcal{E}_{4,1}}{\leq} \max_{\pi} \sum_{i \in \mathcal{H}_1} \left(c\Delta_i^{-2} \log(\tfrac{n}{\pi(i)} \log(\Delta_i^{-2})/\delta) + 5c\Delta_i^{-2} \log(1/\delta)\right)$$
$$\leq \max_{\pi} \sum_{i \in \mathcal{H}_1} c'\Delta_i^{-2} \log(\tfrac{n}{\pi(i)} \log(\Delta_i^{-2})/\delta)$$
$$= \sum_{i=1}^{|\mathcal{H}_1|} c'\Delta^{-2} \log(\tfrac{n}{i} \log(\Delta^{-2})/\delta).$$

The first claimed upper bound of $T$, the diverse means case, is completed by considering the second to last line and noting trivially that $\pi(i) \geq 1$. To obtain the second upper bound of $T$, we consider the last line and note that

$$\sum_{i=1}^{k} \log(\tfrac{n}{i}) \leq \int_0^k \log(\tfrac{n}{x})dx = k\log(n) - (x\log x - x)\Big|_{x=0}^{k} = k\log(\tfrac{n}{k}) + k \leq n.$$

Combining the previous two displays we get

$$\sum_{t=1}^{\infty} \mathbf{1}\{I_t \in \mathcal{H}_1\} \le \sum_{i=1}^{|\mathcal{H}_1|} c\Delta^{-2} \log(\tfrac{n}{i} \log(\Delta^{-2})/\delta)$$

$$\le c|\mathcal{H}_1|\Delta^{-2} \log(\log(\Delta^{-2})/\delta) + \sum_{i=1}^{|\mathcal{H}_1|} c\Delta^{-2} \log(\tfrac{n}{i})$$

$$\le c|\mathcal{H}_1|\Delta^{-2} \log(\log(\Delta^{-2})/\delta) + cn\Delta^{-2}$$

$$\le c''n\Delta^{-2} \log(\log(\Delta^{-2})/\delta).$$

$\square$

# D    Proof of Theorem 3

Our analysis will also make use of the following events, that we will prove each hold with probability at least $1 - \delta$. Let $\beta := \frac{5}{3(1-4\delta)} \log(1/\delta)$ and define:

$$\mathcal{E}_1 := \left\{ \left| \left\{ i \in \mathcal{H}_1 : \bigcap_{t=1}^{\infty}\{|\widehat{\mu}_{i,t} - \mu_i| \le \phi(t, \tfrac{\delta}{\beta})\} \right\} \right| \ge \frac{1}{2}|\mathcal{H}_1| \right\}$$

$$\mathcal{E}_{2,j} := \left\{ \bigcap_{i \in \mathcal{H}_j} \bigcap_{t=1}^{\infty}\{|\widehat{\mu}_{i,t} - \mu_i| \le \phi(t, \tfrac{\delta}{|\mathcal{H}_j|})\} \right\} \quad j \in \{0,1\}$$

Also, recall from Lemma 2 the event

$$\mathcal{E}_3 := \left\{ \forall s \in (0,1] : \sum_{i \in \mathcal{H}_0} \mathbf{1}\{P_{i,*} \le s\} \right.$$

$$\left. \le s|\mathcal{H}_0| + (1 + 4s)\sqrt{2\max\{2s, 2\delta'/n\}|\mathcal{H}_0|\log(\tfrac{\log_2(n/\delta')}{\delta'})} + \tfrac{1+4s}{3}\log(\tfrac{\log_2(n/\delta')}{\delta'}) \right\}$$

Lemma 2 guarantees that this event holds with probability greater than $1 - \delta$.

We restate the theorem using the above events.

**Theorem 3** (FDR, FWPD). *For all $t$ we have $\mathbb{E}[\frac{|\mathcal{S}_t \cap \mathcal{H}_0|}{|\mathcal{S}_t|}] \le \delta$. Moreover, on $\mathcal{E}_1 \cap \mathcal{E}_{2,1} \cap \mathcal{E}_3 \cap \mathcal{E}_{4,0} \cap \mathcal{E}_{4,1}$ (which holds with probability at least $1 - 5\delta$), there exists a $T$ such that*

$$T \lesssim (n - |\mathcal{H}_1|)\Delta^{-2}\log(\max\{|\mathcal{H}_1|, \log\log(n/\delta)\}\log(\Delta^{-2})/\delta) + |\mathcal{H}_1|\Delta^{-2}\log(\log(\Delta^{-2})/\delta)$$

*and $\mathcal{H}_1 \subseteq \mathcal{S}_t$ for all $t \ge T$.*

We need a few technical lemmas, specifically, the proof of events $\mathcal{E}_1$ and $\mathcal{E}_{2,1}$ and their consequences.
**Lemma 4.** $\mathbb{P}(\mathcal{E}_1) \ge 1 - \delta$.

*Proof.* We break the proof up into two cases based on the cardinality $|\mathcal{H}_1|$. If $|\mathcal{H}_1| \le \beta$ then

$$\mathbb{P}\left( \bigcup_{i \in \mathcal{H}_1} \bigcup_{t=1}^{\infty}\{|\widehat{\mu}_{i,t} - \mu_i| \ge \phi(t, \tfrac{\delta}{\beta})\} \right) \le \sum_{i \in \mathcal{H}_1} \mathbb{P}\left( \bigcup_{t=1}^{\infty}\{|\widehat{\mu}_{i,t} - \mu_i| \ge \phi(t, \tfrac{\delta}{\beta})\} \right) \le \delta\frac{|\mathcal{H}_1|}{\beta}$$

which is less than $\delta$ by the case definition. So in what follows, assume that $|\mathcal{H}_1| > \beta$. By definition

$$\mathbb{P}\left( \bigcup_{t=1}^{\infty}\{|\widehat{\mu}_{i,t} - \mu_i| \ge \phi(t, \delta)\} \right) = \mathbb{P}(\rho_i \le \delta) \le \delta.$$

By Bernstein's inequality, with probability at least $1 - \delta$

$$\sum_{i \in \mathcal{H}_1} \mathbf{1}\{\rho_i \le \delta\} \le \delta|\mathcal{H}_1| + \sqrt{2\delta|\mathcal{H}_1|\log(1/\delta)} + \tfrac{1}{3}\log(1/\delta)$$

$$\le \delta|\mathcal{H}_1| + 2\sqrt{\tfrac{1}{2}\delta|\mathcal{H}_1|\log(1/\delta)} + (1 - \tfrac{1}{3})\tfrac{1}{2}\log(1/\delta)$$

$$\le 2\delta|\mathcal{H}_1| + \tfrac{5}{6}\log(1/\delta)$$

where the last line follows from $a + 2\sqrt{ab} + b = (\sqrt{a} + \sqrt{b})^2 \leq 2a + 2b$, which implies

$$\sum_{i \in \mathcal{H}_1} \mathbf{1}\{\rho_i > \delta\} \geq (1 - 2\delta)|\mathcal{H}_1| - \tfrac{5}{6}\log(1/\delta)$$

$$\geq \tfrac{1}{2}|\mathcal{H}_1|$$

where we use the fact that $|\mathcal{H}_1| > \beta = \frac{5}{3(1-4\delta)}\log(1/\delta)$. Combining these two results, and noting that at most one of the cases $|\mathcal{H}_1| > \beta$ or $|H_1| \leq \beta$ can be true, we obtain that $\mathbb{P}(\mathcal{E}_1) \geq 1 - \delta$.  □

**Lemma 5.** $\min\{\mathbb{P}(\mathcal{E}_{2,0}), \mathbb{P}(\mathcal{E}_{2,1})\} \geq 1 - \delta$.

*Proof.* The result follows from a union bound:

$$\mathbb{P}(\mathcal{E}_{2,1}^c) = \mathbb{P}\left(\left\{\bigcup_{i \in \mathcal{H}_1} \bigcup_{t=1}^{\infty}\{|\widehat{\mu}_{i,t} - \mu_i| \leq \phi(t, \tfrac{\delta}{|\mathcal{H}_1|})\}\right\}\right)$$

$$\leq \sum_{i \in \mathcal{H}_1} \mathbb{P}\left(\left\{\bigcup_{t=1}^{\infty}\{|\widehat{\mu}_{i,t} - \mu_i| \leq \phi(t, \tfrac{\delta}{|\mathcal{H}_1|})\}\right\}\right) \leq \sum_{i \in \mathcal{H}_1} \frac{\delta}{|\mathcal{H}_1|} \leq \delta.$$

The proof that $\mathbb{P}(\mathcal{E}_{2,0}) \geq 1 - \delta$ follows analogously.  □

The next lemma shows an important consequence of these events holding.

**Lemma 6.** *If* $\mathcal{E}_1 \cap \mathcal{E}_{2,1}$ *then for all* $t$

$$\mathcal{H}_1 \not\subseteq \mathcal{S}_t \implies \max_{i \in \mathcal{H}_1 \cap \mathcal{S}_t^c} \widehat{\mu}_{i,T_i(t)} + \phi(T_i(t), \tfrac{\delta}{2|\mathcal{S}_t|\vee\beta}) \geq \min_{i \in \mathcal{H}_1} \mu_i.$$

*Proof.* Define the random set $\mathcal{I}_t = \left\{i \in \mathcal{H}_1 : \widehat{\mu}_{i,T_i(t)} + \phi(T_i(t), \tfrac{\delta}{2|\mathcal{S}_t|\vee\beta}) \geq \mu_i\right\}$. We will prove that on $\mathcal{E}_1 \cap \mathcal{E}_{2,1}$ we have $\mathcal{I}_t \cap \mathcal{S}_t^c \neq \emptyset$ which implies the result. First we use the fact that $2|\mathcal{S}_t| \vee \beta \geq \beta$ so that

$$|\mathcal{I}_t| = \left|\left\{i \in \mathcal{H}_1 : \widehat{\mu}_{i,T_i(t)} + \phi(T_i(t), \tfrac{\delta}{2|\mathcal{S}_t|\vee\beta}) \geq \mu_i\right\}\right|$$

$$\geq \left|\left\{i \in \mathcal{H}_1 : \widehat{\mu}_{i,T_i(t)} + \phi(T_i(t), \tfrac{\delta}{\beta}) \geq \mu_i\right\}\right|$$

$$\geq \left|\left\{i \in \mathcal{H}_1 : \bigcap_{t=1}^{\infty}\{|\widehat{\mu}_{i,T_i(t)} - \mu_i| \leq \phi(T_i(t), \tfrac{\delta}{\beta})\}\right\}\right|$$

$$\geq \left|\left\{i \in \mathcal{H}_1 : \bigcap_{t=1}^{\infty}\{|\widehat{\mu}_{i,t} - \mu_i| \leq \phi(t, \tfrac{\delta}{\beta})\}\right\}\right|$$

$$\overset{\mathcal{E}_1}{\geq} \frac{1}{2}|\mathcal{H}_1|.$$

Given $|\mathcal{I}_t| \geq \frac{1}{2}|\mathcal{H}_1|$, if $|\mathcal{S}_t| < \frac{1}{2}|\mathcal{H}_1|$ then $|\mathcal{S}_t| < |\mathcal{I}_t|$ which implies $\mathcal{S}_t^c \cap \mathcal{I}_t \neq \emptyset$. On the other hand, if $|\mathcal{S}_t| \geq |\mathcal{H}_1|/2$ then we use the fact that $2|\mathcal{S}_t| \vee \beta \geq 2|\mathcal{S}_t| \geq |\mathcal{H}_1|$ to observe

$$|\mathcal{I}_t| = \left|\left\{i \in \mathcal{H}_1 : \widehat{\mu}_{i,T_i(t)} + \phi(T_i(t), \tfrac{\delta}{2|\mathcal{S}_t|\vee\beta}) \geq \mu_i\right\}\right|$$

$$\geq \left|\left\{i \in \mathcal{H}_1 : \widehat{\mu}_{i,T_i(t)} + \phi(T_i(t), \tfrac{\delta}{|\mathcal{H}_1|}) \geq \mu_i\right\}\right|$$

$$\geq \left|\left\{i \in \mathcal{H}_1 : \bigcap_{t=1}^{\infty}\{|\widehat{\mu}_{i,T_i(t)} - \mu_i| \leq \phi(T_i(t), \tfrac{\delta}{|\mathcal{H}_1|})\}\right\}\right|$$

$$\geq \left|\left\{i \in \mathcal{H}_1 : \bigcap_{t=1}^{\infty}\{|\widehat{\mu}_{i,t} - \mu_i| \leq \phi(t, \tfrac{\delta}{|\mathcal{H}_1|})\}\right\}\right|$$

$$\overset{\mathcal{E}_{2,1}}{\geq} |\mathcal{H}_1|$$

which implies $\mathcal{I}_t = \mathcal{H}_1$, thus $\mathcal{I}_t \cap \mathcal{S}_t^c = \mathcal{H}_1 \cap \mathcal{S}_t^c$ which is non-empty by assumption.  □

We are now ready to prove Theorem 3.

*Proof.* We proceed similarly to Theorem 1. Note that $\mathcal{S}_t \subseteq \mathcal{S}_{t+1}$ for all $t$ so define $T = \min\{t \in \mathbb{N} : \mathcal{H}_1 \subseteq \mathcal{S}_{t+1}\}$ if such inclusion ever exists, otherwise let $T = \infty$. Then

$$T = \sum_{t=1}^{\infty} \mathbf{1}\{\mathcal{H}_1 \not\subseteq \mathcal{S}_t\}$$

$$= \sum_{t=1}^{\infty} \mathbf{1}\{I_t \in \mathcal{H}_0, \mathcal{H}_1 \not\subseteq \mathcal{S}_t\} + \mathbf{1}\{I_t \in \mathcal{H}_1\}.$$

Note that we are in the TPR setting, like Theorem 2, and so the upperbound $\sum_{t=1}^{\infty} \mathbf{1}\{I_t \in \mathcal{H}_1\} \leq cn\Delta^{-2}\log(\log(\Delta^{-2})/\delta)$ applies here as well. Thus, we only need to bound the first sum.

Lemma 6 (which requires $\mathcal{E}_1 \cap \mathcal{E}_{2,1}$) says that if there is at least one arm from $\mathcal{H}_1$ not in $\mathcal{S}_t$ then the largest upper confidence bound of some arm in $\mathcal{H}_1 \cap \mathcal{S}_t^c$ is *at least* as large as $\min_{j \in \mathcal{H}_1} \mu_j \geq \mu_0 + \Delta$. Thus, $\{\mathcal{H}_1 \not\subseteq \mathcal{S}_t\}$ implies that

$$\arg\max_{i \in \mathcal{S}_t^c} \widehat{\mu}_{i,T_i(t)} + \phi(T_i(t), \delta) \geq \min_{i \in \mathcal{H}_1} \mu_i = \mu_0 + \Delta.$$

On the other hand, for $\kappa = \frac{1}{1-2\delta'(1+4\delta')}|\mathcal{H}_1| + \frac{4(1+4\delta')/3}{1-2\delta'(1+4\delta')}\log(\frac{5\log_2(n/\delta')}{\delta'})$, we have $|\mathcal{S}_t| \leq \kappa$ from Lemma 2 (which requires $\mathcal{E}_3$), and for any $i \in \mathcal{H}_0$ we have

$$\widehat{\mu}_{i,T_i(t)} + \phi(T_i(t), \tfrac{\delta}{|\mathcal{S}_t|\vee\beta}) \leq \mu_i + \phi(T_i(t), \tfrac{\delta}{|\mathcal{S}_t|\vee\beta}) + \phi(T_i(t), \rho_i)$$

$$\leq \mu_i + \phi(T_i(t), \tfrac{\delta}{\kappa}) + \phi(T_i(t), \rho_i)$$

$$\leq \mu_i + 2\phi(T_i(t), \tfrac{\delta\rho_i}{\kappa})$$

$$\leq \mu_0 + 2\phi(T_i(t), \tfrac{\delta\rho_i}{\kappa})$$

so that $\widehat{\mu}_{i,T_i(t)} + \phi(T_i(t), \tfrac{\delta}{|\mathcal{S}_t|\vee\beta}) \leq \mu_0 + \Delta$ whenever $T_i(t) \geq \phi^{-2}(\tfrac{\Delta}{2}, \tfrac{\delta\rho_i}{\kappa})$. Thus,

$$\sum_{t=1}^{\infty} \mathbf{1}\{I_t \in \mathcal{H}_0, \mathcal{H}_1 \not\subseteq \mathcal{S}_t\} \leq \sum_{i \in \mathcal{H}_0} \phi^{-2}(\tfrac{\Delta}{2}, \tfrac{\delta\rho_i}{\kappa})$$

$$\leq \sum_{i \in \mathcal{H}_0} c\Delta^{-2}\log(\kappa\log(\Delta^{-2})/\delta) + c\Delta^{-2}\log(1/\rho_i)$$

$$\overset{\mathcal{E}_{4,0}}{\leq} \sum_{i \in \mathcal{H}_0} c'\Delta^{-2}\log(\max\{|\mathcal{H}_1|, \log\log(n/\delta)\}\log(\Delta^{-2})/\delta) + 5c\Delta^{-2}\log(1/\delta)$$

$$\leq |\mathcal{H}_0|c''\Delta^{-2}\log(\max\{|\mathcal{H}_1|, \log\log(n/\delta)\}\log(\Delta^{-2})/\delta)$$

We conclude that

$$T = \sum_{t=1}^{T} \mathbf{1}\{I_t \in \mathcal{H}_0, \mathcal{H}_1 \not\subseteq \mathcal{S}_t\} + \mathbf{1}\{I_t \in \mathcal{H}_1, \mathcal{H}_1 \not\subseteq \mathcal{S}_t\}$$

$$\leq (n - |\mathcal{H}_1|)c\Delta^{-2}\log(\max\{|\mathcal{H}_1|, \log\log(n/\delta)\}\log(\Delta^{-2})/\delta) + cn\Delta^{-2}\log(\log(\Delta^{-2})/\delta)$$

$$\leq (n - |\mathcal{H}_1|)c'\Delta^{-2}\log(\max\{|\mathcal{H}_1|, \log\log(n/\delta)\}\log(\Delta^{-2})/\delta) + c'|\mathcal{H}_1|\Delta^{-2}\log(\log(\Delta^{-2})/\delta).$$

$\square$

# E    Proof of Theorem 4

We restate the theorem using the above events.

**Theorem 4** (FWER, FWPD). *For all $t$ we have $\mathbb{E}[\frac{|\mathcal{S}_t \cap \mathcal{H}_0|}{|\mathcal{S}_t|}] \leq \delta$. Moreover, on $\mathcal{E}_1 \cap \mathcal{E}_{2,0} \cap \mathcal{E}_{2,1} \cap \mathcal{E}_3 \cap \mathcal{E}_{4,0} \cap \mathcal{E}_{4,1}$ (which holds with probability at least $1 - 6\delta$), we have $\mathcal{H}_0 \cap \mathcal{R}_t = \emptyset$ for all $t \in \mathbb{N}$ and there exists a $T$ such that*

$$T \lesssim (n - |\mathcal{H}_1|)\Delta^{-2}\log(\max\{|\mathcal{H}_1|, \log\log(n/\delta)\}\log(\Delta^{-2})/\delta)$$

$$+ |\mathcal{H}_1|\Delta^{-2}\log(\max\{n - (1 - 2\delta(1+4\delta))|\mathcal{H}_1|, \log\log(n/\delta)\}\log(\Delta^{-2})/\delta)$$

*and $\mathcal{H}_1 \subseteq \mathcal{R}_t$ for all $t \geq T$. Note, together this implies $\mathcal{H}_1 = \mathcal{R}_t$ for all $t \geq T$.*

The following lemma shows that a tight control on the size of $\mathcal{S}_t$ allows us to conclude a FWER.

**Lemma 7.** *If $\mathcal{E}_3 \cap \mathcal{E}_{2,0}$ holds then $\mathcal{R}_t \cap \mathcal{H}_0 = \emptyset$ for all t.*

*Proof.* By Lemma 2 (which requires $\mathcal{E}_3$) we have $|\mathcal{S}_t| \leq \frac{1}{1-2\delta'(1+4\delta')}|\mathcal{H}_1| + \frac{4(1+4\delta')/3}{1-2\delta'(1+4\delta')} \log(\frac{5\log_2(n/\delta')}{\delta'}) = \frac{|\mathcal{H}_1|+\eta}{1-2\delta'(1+4\delta')}$ for all times $t$ and $\eta = \frac{4(1+4\delta')}{3} \log(5\log_2(n)/\delta')$. This implies

$$n - (1 - 2\delta'(1+4\delta'))|\mathcal{S}_t| + \eta \geq n - |\mathcal{H}_1| = |\mathcal{H}_0| \tag{3}$$

but for $i \in \mathcal{S}_t \cap \mathcal{H}_0 \subseteq \mathcal{H}_0$ we have that $\mathcal{E}_{2,0}$ applies so

$$\widehat{\mu}_{i,T_i(t)} - \phi(T_i(t), \tfrac{\delta}{n-(1-2\delta'(1+4\delta'))|\mathcal{S}_t|+\eta}) \leq \widehat{\mu}_{i,T_i(t)} - \phi(T_i(t), \tfrac{\delta}{|\mathcal{H}_0|}) \overset{\mathcal{E}_{2,0}}{\leq} \mu_i \leq \mu_0$$

where the last inequality holds because $\max_{i \in \mathcal{H}_0} \mu_i \leq \mu_0$. Thus, no arms from $\mathcal{H}_0$ will be added to $\mathcal{R}_t$. $\qquad\square$

Now that we have FWER control, we need to show that all the arms in $\mathcal{H}_1$ are added to $\mathcal{R}_t$ in the claimed amount of time.

*Proof.* Note that $\mathcal{R}_t \subseteq \mathcal{R}_{t+1}$ for all $t$ so define $T = \min\{t \in \mathbb{N} : \mathcal{H}_1 \subseteq \mathcal{R}_{t+1}\}$ if such inclusion ever exists, otherwise let $T = \infty$. Noting that $\mathcal{R}_t \subseteq \mathcal{S}_t$ we have

$$T = \sum_{t=1}^{\infty} \mathbf{1}\{\mathcal{H}_1 \not\subseteq \mathcal{R}_t\}$$

$$= \sum_{t=1}^{\infty} \mathbf{1}\{\mathcal{H}_1 \not\subseteq \mathcal{S}_t\} + \mathbf{1}\{\mathcal{H}_1 \not\subseteq \mathcal{R}_t, \mathcal{H}_1 \subseteq \mathcal{S}_t\}$$

$$= \sum_{t=1}^{\infty} \mathbf{1}\{\mathcal{H}_1 \not\subseteq \mathcal{S}_t\} + \mathbf{1}\{J_t \in \mathcal{H}_0, \mathcal{H}_1 \not\subseteq \mathcal{R}_t, \mathcal{H}_1 \subseteq \mathcal{S}_t\} + \mathbf{1}\{J_t \in \mathcal{H}_1, \mathcal{H}_1 \not\subseteq \mathcal{R}_t, \mathcal{H}_1 \subseteq \mathcal{S}_t\}$$

**First sum.** Note that we are in the FWPD setting, so the selection rule for $I_t$ identical to that of the setting of Theorem 3, and $\sum_{t=1}^{\infty} \mathbf{1}\{\mathcal{H}_1 \not\subseteq \mathcal{S}_t\}$ is precisely what is bounded in the proof of Theorem 3 (which requires $\mathcal{E}_1 \cap \mathcal{E}_{2,1} \cap \mathcal{E}_3 \cap \mathcal{E}_{4,0} \cap \mathcal{E}_{4,1}$). Thus,

$$\sum_{t=1}^{T} \mathbf{1}\{\mathcal{H}_1 \not\subseteq \mathcal{S}_t\}$$
$$\leq (n - |\mathcal{H}_1|)c'\Delta^{-2}\log(\max\{|\mathcal{H}_1|, \log\log(n/\delta)\}\log(\Delta^{-2})/\delta) + c'|\mathcal{H}_1|\Delta^{-2}\log(\log(\Delta^{-2})/\delta).$$

**Second sum.** Recall that $J_t = \arg\min_{i \in \mathcal{S}_t \setminus \mathcal{R}_t} \widehat{\mu}_{i,T_i(t)} + \phi(T_i(t), \tfrac{\delta}{|\mathcal{S}_t|})$. Now, on the event $\{\mathcal{H}_1 \not\subseteq \mathcal{R}_t, \mathcal{H}_1 \subseteq \mathcal{S}_t\}$ we have that $|\mathcal{H}_1| \leq |\mathcal{S}_t|$ and that there exists a $j \in \mathcal{H}_1 \cap (\mathcal{S}_t \setminus \mathcal{R}_t)$ such that

$$\widehat{\mu}_{j,T_j(t)} + \phi(T_j(t), \tfrac{\delta}{|\mathcal{S}_t|}) \geq \widehat{\mu}_{j,T_j(t)} + \phi(T_j(t), \tfrac{\delta}{|\mathcal{H}_1|})$$
$$\overset{\mathcal{E}_{2,1}}{\geq} \mu_j$$
$$\geq \mu_0 + \Delta.$$

On the other hand, for $\kappa = \frac{1}{1-2\delta'(1+4\delta')}|\mathcal{H}_1| + \frac{4(1+4\delta')/3}{1-2\delta'(1+4\delta')} \log(\frac{5\log_2(n/\delta')}{\delta'})$, we have $|\mathcal{S}_t| \leq \kappa$ from Lemma 2 (which requires $\mathcal{E}_3$), and for any $i \in \mathcal{H}_0 \cap (\mathcal{S}_t \setminus \mathcal{R}_t)$ we have

$$\widehat{\mu}_{i,T_i(t)} + \phi(T_i(t), \tfrac{\delta}{|\mathcal{S}_t|}) \leq \mu_i + \phi(T_i(t), \tfrac{\delta}{\kappa}) + \phi(T_i(t), \rho_i)$$
$$\leq \mu_i + 2\phi(T_i(t), \tfrac{\delta\rho_i}{\kappa})$$
$$\leq \mu_0 + 2\phi(T_i(t), \tfrac{\delta\rho_i}{\kappa})$$

so that $\widehat{\mu}_{i,T_i(t)} + \phi(T_i(t), \frac{\delta}{|\mathcal{S}_t|}) \leq \mu_0 + \Delta$ whenever $T_i(t) \geq \phi^{-2}(\frac{\Delta}{2}, \frac{\delta\rho_i}{\kappa})$. By an identical argument to that made in the proof of Theorem 3 we very crudely have the bound

$$\sum_{t=1}^{\infty} \mathbf{1}\{J_t \in \mathcal{H}_0, \mathcal{H}_1 \not\subseteq \mathcal{R}_t, \mathcal{H}_1 \subseteq \mathcal{S}_t\} \leq \sum_{i \in \mathcal{H}_0} \phi^{-2}(\tfrac{\Delta}{2}, \tfrac{\delta\rho_i}{\kappa})$$

$$\leq c|\mathcal{H}_0|\Delta^{-2}\log(\max\{|\mathcal{H}_1|, \log\log(n/\delta)\}\log(\Delta^{-2})/\delta).$$

**Third sum.** An arm $j \in \mathcal{H}_1 \cap (\mathcal{S}_t \setminus \mathcal{R}_t)$ is accepted into $\mathcal{R}_t$ if $\widehat{\mu}_{i,T_i(t)} - \phi(T_i(t), \frac{\delta}{\chi_t}) \geq \mu_0$ where $\chi_t = n - (1 - 2\delta'(1 + 4\delta'))|\mathcal{S}_t| + \frac{4(1+4\delta')}{3}\log(5\log_2(n/\delta')/\delta')$. On the event $\{\mathcal{H}_1 \not\subseteq \mathcal{R}_t, \mathcal{H}_1 \subseteq \mathcal{S}_t\}$ we have $\chi_t \leq n - (1 - 2\delta'(1 + 4\delta'))|\mathcal{H}_1| + \frac{4(1+4\delta')}{3}\log(5\log_2(n/\delta')/\delta') =: u$. Thus, for $j \in \mathcal{H}_1 \cap (\mathcal{S}_t \setminus \mathcal{R}_t)$

$$\widehat{\mu}_{j,T_j(t)} - \phi(T_j(t), \tfrac{\delta}{\chi_t}) \geq \widehat{\mu}_{j,T_j(t)} - \phi(T_j(t), \tfrac{\delta}{u})$$

$$\geq \mu_j - \phi(T_j(t), \tfrac{\delta}{u}) - \phi(T_j(t), \rho_j)$$

$$\geq \mu_j - 2\phi(T_j(t), \tfrac{\delta\rho_j}{u})$$

$$\geq \mu_0 + \Delta - 2\phi(T_j(t), \tfrac{\delta\rho_j}{u})$$

so that $\widehat{\mu}_{j,T_j(t)} - \phi(T_j(t), \frac{\delta}{\chi_t}) \geq \mu_0$ whenever $T_j(t) \geq \phi^{-1}(\frac{\Delta}{2}, \frac{\delta\rho_j}{u})$. By the same arguments made throughout these proofs we have

$$\sum_{t=1}^{\infty} \mathbf{1}\{J_t \in \mathcal{H}_1, \mathcal{H}_1 \not\subseteq \mathcal{R}_t, \mathcal{H}_1 \subseteq \mathcal{S}_t\} \leq \sum_{j \in \mathcal{H}_1} \phi^{-1}(\tfrac{\Delta}{2}, \tfrac{\delta\rho_j}{u})$$

$$\leq c|\mathcal{H}_1|\Delta^{-2}\log(u\log(\Delta^{-2})/\delta)$$

$$\leq c'|\mathcal{H}_1|\Delta^{-2}\log(\max\{n - (1 - 2\delta(1 + 4\delta))|\mathcal{H}_1|, \log\log(n/\delta)\}\log(\Delta^{-2})/\delta).$$

Summing all three sums together yields the result. $\qquad\square$

# F   Proof of Theorem 5

We define a new event,

$$\mathcal{E}_5 := \left\{\sum_{i \in \mathcal{H}_1} \mathbf{1}\{\rho_i \leq \delta\} \leq \delta|\mathcal{H}_1| + \sqrt{2\delta|\mathcal{H}_1|\log(1/\delta)} + \tfrac{1}{3}\log(1/\delta)\right\}$$

By a direct application of Bernstein's inequality, this holds with probability greater than $1 - \delta$.

**Theorem 5** (FWER, TPR). *For all $t$ we have $\mathbb{E}[\frac{|\mathcal{S}_t \cap \mathcal{H}_0|}{|\mathcal{S}_t|}] \leq \delta$. Moreover, on $\mathcal{E}_1 \cap \mathcal{E}_{2,1} \cap \mathcal{E}_3 \cap \mathcal{E}_{4,0} \cap \mathcal{E}_{4,1} \cap \mathcal{E}_{2,0} \cap \mathcal{E}_5$ (which holds with probability at least $1 - 7\delta$), we have $\mathcal{H}_0 \cap \mathcal{R}_t = \emptyset$ for all $t \in \mathbb{N}$ and there exists a $T$ such that*

$$T \lesssim (n - |\mathcal{H}_1|)\Delta^{-2}\log(\log(\Delta^{-2})/\delta)$$

$$+ |\mathcal{H}_1|\Delta^{-2}\log(\max\{n - (1 - 2\delta)|\mathcal{H}_1|, \log\log(n/\delta)\}\log(\Delta^{-2})/\delta)$$

*and $\mathbb{E}[\frac{|\mathcal{R}_t \cap \mathcal{H}_1|}{|\mathcal{H}_1|}] \geq 1 - \delta$ for all $t \geq T$.*

*Proof.* First, we invoke Lemma 7 (which requires $\mathcal{E}_3 \cap \mathcal{E}_{2,0}$) which controls the FWER. All that is left is to control the sample complexity.

Let $\mathcal{I} = \{i \in \mathcal{H}_1 : \rho_i \geq \delta\}$ be the same random set defined in the proof of Theorem 2. Note that $\mathcal{R}_t \subseteq \mathcal{R}_{t+1}$ for all $t$ so define $T = \min\{t \in \mathbb{N} : \mathcal{I} \subseteq \mathcal{R}_{t+1}\}$ if such inclusion ever exists, otherwise

let $T = \infty$. Noting that $\mathcal{R}_t \subseteq \mathcal{S}_t$ we have

$$
\begin{aligned}
T &= \sum_{t=1}^{\infty} \mathbf{1}\{\mathcal{I} \not\subseteq \mathcal{R}_t\} \\
&= \sum_{t=1}^{\infty} \mathbf{1}\{\mathcal{I} \not\subseteq \mathcal{S}_t\} + \mathbf{1}\{\mathcal{I} \not\subseteq \mathcal{R}_t, \mathcal{I} \subseteq \mathcal{S}_t\} \\
&= \sum_{t=1}^{\infty} \mathbf{1}\{\mathcal{I} \not\subseteq \mathcal{S}_t\} + \mathbf{1}\{J_t \in \mathcal{H}_0, \mathcal{I} \not\subseteq \mathcal{R}_t, \mathcal{I} \subseteq \mathcal{S}_t\} + \mathbf{1}\{J_t \in \mathcal{H}_1, \mathcal{I} \not\subseteq \mathcal{R}_t, \mathcal{I} \subseteq \mathcal{S}_t\}.
\end{aligned}
$$

**First sum.** Since we are in the TPR setting we have $\xi_t = 1$ so the first sum is precisely the quantity bounded in the proof of Theorem 2. Thus,

$$
\sum_{t=1}^{\infty} \mathbf{1}\{\mathcal{I} \not\subseteq \mathcal{S}_t\} \le cn\Delta^{-2} \log(\log(\Delta^{-2})/\delta).
$$

**Second sum.** Recall that $J_t = \arg\max_{i \in \mathcal{S}_t \setminus \mathcal{R}_t} \widehat{\mu}_{i, T_i(t)} + \phi(T_i(t), \delta)$. Now, on the event $\{\mathcal{I} \not\subseteq \mathcal{R}_t, \mathcal{I} \subseteq \mathcal{S}_t\}$ there exists a $j \in \mathcal{I} \cap (\mathcal{S}_t \setminus \mathcal{R}_t)$ such that

$$
\begin{aligned}
\widehat{\mu}_{j, T_j(t)} + \phi(T_j(t), \delta) &\ge \mu_j + \phi(T_j(t), \delta) - \phi(T_j(t), \rho_j) \\
&\ge \mu_j \\
&\ge \mu_0 + \Delta
\end{aligned}
$$

where the first inequality follows by the definition of $\mathcal{I}$. On the other hand, for any $i \in \mathcal{H}_0$ we have

$$
\begin{aligned}
\widehat{\mu}_{i, T_i(t)} + \phi(T_i(t), \delta) &\le \mu_i + \phi(T_i(t), \delta) + \phi(T_i(t), \rho_i) \\
&\le \mu_i + 2\phi(T_i(t), \delta\rho_i) \\
&\le \mu_0 + 2\phi(T_i(t), \delta\rho_i)
\end{aligned}
$$

so that $\widehat{\mu}_{i, T_i(t)} + \phi(T_i(t), \delta) \le \mu_0 + \Delta$ whenever $T_i(t) \ge \phi^{-1}(\frac{\Delta}{2}, \delta\rho_i)$. Thus, by the same sequence of steps used in Theorem 2 we have

$$
\sum_{t=1}^{\infty} \mathbf{1}\{J_t \in \mathcal{H}_0, \mathcal{I} \not\subseteq \mathcal{R}_t, \mathcal{I} \subseteq \mathcal{S}_t\} \le \sum_{i \in \mathcal{H}_0} \phi^{-1}(\tfrac{\Delta}{2}, \delta\rho_i) \overset{\mathcal{E}_{4,0}}{\le} c|\mathcal{H}_0|\Delta^{-2} \log(\log(\Delta^{-2})/\delta).
$$

**Third sum.** In bounding the analogous sum in the proof of Theorem 4 we used the fact that $\mathcal{H}_1 \subseteq \mathcal{S}_t$ to lowerbound $|\mathcal{S}_t|$ to obtain an upperbound on $\xi_t$. Now that we only have $\mathcal{I} \subseteq \mathcal{S}_t$ we observe that

$$
|\mathcal{S}_t| \ge |\mathcal{I}| \ge |\mathcal{H}_1| - \sum_{i \in \mathcal{H}_1} \mathbf{1}\{\rho_i \le \delta\} \overset{\mathcal{E}_5}{\ge} |\mathcal{H}_1|(1 - \delta - \sqrt{\tfrac{2\delta \log(1/\delta)}{|\mathcal{H}_1|}} - \tfrac{\log(1/\delta)}{3|\mathcal{H}_1|}).
$$

Using this approximation the same argument yields

$$
\begin{aligned}
&\sum_{t=1}^{\infty} \mathbf{1}\{J_t \in \mathcal{H}_1, \mathcal{I} \not\subseteq \mathcal{R}_t, \mathcal{I} \subseteq \mathcal{S}_t\} \\
&\le c'|\mathcal{H}_1|\Delta^{-2} \log(\max\{n - (1 - 3\delta - \sqrt{\tfrac{2\delta \log(1/\delta)}{|\mathcal{H}_1|}} - \tfrac{\log(1/\delta)}{3|\mathcal{H}_1|})|\mathcal{H}_1|, \log\log(n/\delta)\} \log(\Delta^{-2})/\delta) \\
&\le c''|\mathcal{H}_1|\Delta^{-2} \log(\max\{n - (1 - 3\delta - \sqrt{2\delta \log(1/\delta)/|\mathcal{H}_1|})|\mathcal{H}_1|, \log\log(n/\delta)\} \log(\Delta^{-2})/\delta).
\end{aligned}
$$

$\square$

# G  Successive Elimination and Uniform Allocation Algorithms

The following gives a sample complexity bound for Successive Elimination and Uniform allocation strategies. Note that for these algorithms, up to $n$ arms are pulled at each time $t$.

**Algorithm 2** Uniform and Successive elimination algorithms for identifying arms with means above $\mu_0$.

---

**Input:** Threshold $\mu_0$, confidence $\delta$, confidence interval $\phi(\cdot, \cdot)$
**Initialize:** Set $\mathcal{S}_1 = \emptyset$
**For** $t = 1, 2, \ldots$
    **if Successive Elimination**
        **Pull each and every arm** in $[n] - \mathcal{S}_t$.
    **else if Uniform**
        **Pull each and every arm** in $[n]$.
    **Apply** Benjamini-Hochberg [11] selection to obtain FDR-controlled set $\mathcal{S}_t$:
        $s(k) = \{ i \in [n] \setminus \mathcal{S}_t : \widehat{\mu}_{i,t} - \phi(t, \delta\frac{k}{n}) \geq \mu_0 \}, \forall k \in [n]$
        $\mathcal{S}_{t+1} = \mathcal{S}_t \cup s(\widehat{k})$ where $\widehat{k} = \max\{k \in [n] : |s(k)| \geq k\}$ (if $\nexists \widehat{k}$ set $\mathcal{S}_{t+1} = \mathcal{S}_t$)

---

**Theorem 6.** *For both the Successive Elimination and Uniform Allocation algorithms, for all $t$ we have that $\mathbb{E}[\frac{|\mathcal{S}_t \cap \mathcal{H}_0|}{|\mathcal{S}_t|}] \leq \delta$. In addition, in the case of successive elimination, if $\eta = \delta + \sqrt{\frac{2\delta \log(1/\delta)}{|\mathcal{H}_1|}} + \frac{\log(1/\delta)}{3|\mathcal{H}_1|}$, then on the event $\mathcal{E}_5$ (which holds with probability greater than $1 - \delta$) then there exists a $T$ such that*

$$T \lesssim \min \Big\{ n\Delta^{-2} \log(\tfrac{n}{(1-\eta)|\mathcal{H}_1|} \log(\Delta^{-2})/\delta),$$

$$(n - (1-\eta)|\mathcal{H}_1|)\Delta^{-2} \log(\tfrac{n}{(1-\eta)|\mathcal{H}_1|} \log(\Delta^{-2})/\delta) + \sum_{i \in \mathcal{H}_1} \Delta_i^{-2} \log(n \log(\Delta_i^{-2})/\delta) \Big\}$$

*and in the case of Uniform allocation,*

$$T \lesssim n\Delta^{-2} \log(\tfrac{n}{(1-\eta)|\mathcal{H}_1|} \log(\Delta^{-2})/\delta)$$

*and $\mathbb{E}[\frac{|\mathcal{S}_t \cap \mathcal{H}_1|}{|\mathcal{S}_t|}] > 1 - \delta$ for all $t \geq T$.*

*Proof.* Define the random set $\mathcal{I} = \{i \in \mathcal{H}_1 : \rho_i \geq \delta\}$. Note, this is equivalent to $\mathcal{I} = \{i \in \mathcal{H}_1 : \widehat{\mu}_{i,T_i(t)} + \phi(T_i(t), \delta) \geq \mu_i \; \forall t \in \mathbb{N}\}$ since if $\rho_i \geq \delta$ then $\phi(T_i(t), \delta) > \phi(T_i(t), \rho_i)$ and

$$\widehat{\mu}_{i,T_i(t)} + \phi(T_i(t), \delta) \geq \mu_i + \phi(T_i(t), \delta) - \phi(T_i(t), \rho_i) \geq \mu_i.$$

Our aim is to show that this "well-behaved" set of indices $\mathcal{I}$ will be added to $\mathcal{S}_t$ in the claimed amount of time. This is sufficient for the TPR result because $\mathbb{E}|\mathcal{I}| = \sum_{i \in \mathcal{H}_1} \mathbb{E}[\mathbf{1}\{\rho_i > \delta\}] = \sum_{i \in \mathcal{H}_1} \mathbb{P}(\rho_i > \delta) \geq (1 - \delta)|\mathcal{H}_1|$ since each $\rho_i$ is a sub-uniformly distributed random variable. To be clear, we are not claiming which *particular* indices of $\mathcal{H}_1$ will be added to $\mathcal{S}_t$ (indeed, $\mathcal{I}$ is a random set), just that their number exceeds $(1 - \delta)|\mathcal{H}_1|$ in expectation.

First we consider the case of Successive Elimination. Let $T_i = \min\{t \in \mathbb{N} : i \notin \mathcal{S}_t\}$ be the random number of times arm $i$ is pulled until the last time when it is added to $\mathcal{S}_t$ (may possibly be infinite). At round $t$ all arms in $[n] - \mathcal{S}_t$ are pulled and once an arm is added to $\mathcal{S}_t$ it will never be pulled again. Note that $i \in s(k)$ implies $\widehat{\mu}_{i,t} - \phi(t, \delta\frac{k}{n}) \geq \mu_0$. Since for all $i \in \mathcal{I}$

$$\widehat{\mu}_{i,t} - \phi(t, \delta\tfrac{k}{n}) \geq \mu_i - \phi(t, \delta\tfrac{k}{n}) - \phi(t, \delta)$$
$$\geq \mu_i - 2\phi(t, \delta\tfrac{k}{n})$$
$$= \mu_0 + \Delta_i - 2\phi(t, \delta\tfrac{k}{n})$$

we have that $i \in s(k)$ whenever $t \geq \phi^{-1}(\frac{\Delta_i}{2}, \delta\frac{k}{n})$. In particular, because $i \in s(1)$ implies $i \in \mathcal{S}_t$ we have that $T_i \leq \phi^{-1}(\frac{\Delta_i}{2}, \delta\frac{1}{n})$ for all $i \in \mathcal{I}$. But if $\Delta = \min_{i \in \mathcal{H}_1} \Delta_i$ then we also have necessarily that $t \leq \phi^{-1}(\frac{\Delta}{2}, \delta\frac{|\mathcal{I}|}{n})$ since at this time, all arms in $\mathcal{I}$ will be in $s(|\mathcal{I}|)$, which means $\widehat{k} \geq |\mathcal{I}|$ and so $\mathcal{I} \subseteq \mathcal{S}_t$. This, of course, implies $T_i \leq \phi^{-1}(\frac{\Delta}{2}, \delta\frac{|\mathcal{I}|}{n})$ for all $i \in [n] \setminus \mathcal{I}$. Putting these pieces together,

we have that

$$\sum_{i=1}^{n} T_i \leq \min\left\{ n\phi^{-1}(\tfrac{\Delta}{2}, \delta\tfrac{|\mathcal{I}|}{n}), (n-|\mathcal{I}|)\phi^{-1}(\tfrac{\Delta}{2}, \delta\tfrac{|\mathcal{I}|}{n}) + \sum_{i\in\mathcal{I}} \phi^{-1}(\tfrac{\Delta_i}{2}, \delta\tfrac{1}{n}) \right\}$$

$$\leq \min\left\{ cn\Delta^{-2}\log(\tfrac{n}{|\mathcal{I}|}\log(\Delta^{-2})/\delta), \right.$$

$$\left. c(n-|\mathcal{I}|)\Delta^{-2}\log(\tfrac{n}{|\mathcal{I}|}\log(\Delta^{-2})/\delta) + \sum_{i\in\mathcal{I}} c\Delta_i^{-2}\log(n\log(\Delta_i^{-2})/\delta) \right\}.$$

Now on event $\mathcal{E}_5$ (which holds with probability at least $1-\delta$) we have

$$|\mathcal{I}| \geq (1 - \delta - \sqrt{\tfrac{2\delta\log(1/\delta)}{|\mathcal{H}_1|}} - \tfrac{\log(1/\delta)}{3|\mathcal{H}_1|})|\mathcal{H}_1|$$

which implies that for $\eta = \delta + \sqrt{\tfrac{2\delta\log(1/\delta)}{|\mathcal{H}_1|}} + \tfrac{\log(1/\delta)}{3|\mathcal{H}_1|}$ we have

$$\sum_{i=1}^{n} T_i \leq \min\left\{ cn\Delta^{-2}\log(\tfrac{n}{(1-\eta)|\mathcal{H}_1|}\log(\Delta^{-2})/\delta), \right.$$

$$\left. c(n-(1-\eta)|\mathcal{H}_1|)\Delta^{-2}\log(\tfrac{n}{(1-\eta)|\mathcal{H}_1|}\log(\Delta^{-2})/\delta) + \sum_{i\in\mathcal{H}_1} c\Delta_i^{-2}\log(n\log(\Delta_i^{-2})/\delta) \right\}.$$

Now, in the case of Uniform allocation, we never stop pulling arms once they enter $\mathcal{S}_t$. Hence, we need to consider the number of samples needed before all the arms in $\mathcal{I} \subset \mathcal{S}_t$. By the same reasoning as above, this is bounded by $cn\Delta^{-2}\log(\tfrac{n}{(1-\eta)|\mathcal{H}_1|}\log(\Delta^{-2})/\delta)$. $\qquad\square$

The following two theorems provide lower bounds for Uniform allocation.

**Theorem 7** (FDR, TPR). *Fix $\delta < 1/40$, $\Delta > 0$. and $k < n/2$. For any $\mathcal{H}_1 \subseteq [n]$ such that $|\mathcal{H}_1| = k$ define an instance $(\{\nu_i\}_{i=1}^n, \mu_0)$ with $\mu_0 = 0$, $\nu_i = \mathcal{N}(\Delta, 1)$ for $i \in \mathcal{H}_1$ , $\nu_i = \mathcal{N}(0, 1)$ for $i \in \mathcal{H}_0$. Any algorithm that samples each arm an equal number of times before outputing a set $\mathcal{S} \subseteq [n]$ after $\tau$ total samples, and is FDR-$\delta$ and TPR-$\delta$, $\tau$ on $(\{\nu_i\}_{i=1}^n, \mu_0)$ for all $\mathcal{H}_1 \subseteq [n]$ such that $|\mathcal{H}_1| = k$ simultaneously, must satisfy $t \gtrsim n\Delta^{-2}\log(n/k)$.*

*Proof.* The proof is based on the construction of [29] which states that for any $n > 2k$, there exists a collection $\mathcal{M}_{n,k}$ of subsets of $[n]$ where a) each $\pi \in \mathcal{M}_{n,k}$ has weight $|\pi| = k$, b) $2k \geq |\pi \triangle \pi'| > k$ for all $\pi \neq \pi' \in \mathcal{M}_{n,k}$, and c) $|\mathcal{M}_{n,k}| \geq (\tfrac{n}{6k})^{k/4}$. Each $\pi$ gives rise to an instance of the problem $(\{\nu_i\}_{i=1}^n, \mu_0)$ where $\mu_0 = 0$, $\mathcal{H}_1 = \pi$ and so $\nu_i = \mathcal{N}(\Delta, 1)$ if $i \in \pi$, otherwise and $\nu_i = \mathcal{N}(\mu_0, 1)$ otherwise. In particular, for each instance $|\mathcal{H}_1| = k$.

Note that if $(i)$, $\frac{|\mathcal{H}_1 \cap \mathcal{S}_t^c|}{|\mathcal{H}_1|} = 1 - \frac{|\mathcal{H}_1 \cap \mathcal{S}_t|}{|\mathcal{H}_1|} \leq \eta$ and $(ii)$ $\frac{|\mathcal{H}_0 \cap \mathcal{S}_t|}{|\mathcal{H}_0 \cap \mathcal{S}_t| + |\mathcal{H}_1 \cap \mathcal{S}_t|} = \frac{|\mathcal{H}_0 \cap \mathcal{S}_t|}{|\mathcal{S}_t|} \leq \eta$ then

$$|\mathcal{H}_1 \triangle \mathcal{S}_t| = |\mathcal{H}_1 \cap \mathcal{S}_t^c| + |\mathcal{H}_1^c \cap \mathcal{S}_t|$$

$$= |\mathcal{H}_1 \cap \mathcal{S}_t^c| + |\mathcal{H}_0 \cap \mathcal{S}_t|$$

$$\overset{(i)}{\leq} \eta|\mathcal{H}_1| + |\mathcal{H}_0 \cap \mathcal{S}_t|$$

$$\overset{(ii)}{\leq} \eta|\mathcal{H}_1| + \tfrac{\eta}{1-\eta}|\mathcal{H}_1 \cap \mathcal{S}_t|$$

$$\leq \tfrac{2\eta}{1-\eta}|\mathcal{H}_1|.$$

Thus, if $(i)$ and $(ii)$ hold and $\eta < 1/5$ then $|\mathcal{H}_1 \triangle \mathcal{S}_t| \leq k/2$ and so $\mathcal{H}_1 \triangle \mathcal{S}_t$ can therefore be used as an estimator for any $\mathcal{H}_1 = \pi \in \mathcal{M}_{n,k}$ since $\min_{\pi, \pi' \in \mathcal{M}_{n,k}} |\pi \triangle \pi'| > k$.

By assumption, $\mathbb{E}[\frac{|\mathcal{H}_0 \cap \mathcal{S}_t|}{|\mathcal{S}_t|}] \leq \delta$ so by Markov's inequality we have $\mathbb{P}(\frac{|\mathcal{H}_0 \cap \mathcal{S}_t|}{|\mathcal{S}_t|} \geq 8\delta) \leq 1/8$. Likewise, by assumption $\mathbb{E}[\frac{|\mathcal{H}_1 \cap \mathcal{S}_t^c|}{|\mathcal{H}_1|}] = 1 - \mathbb{E}[\frac{|\mathcal{H}_1 \cap \mathcal{S}_t|}{|\mathcal{H}_1|}] \leq \delta$, so again $\mathbb{P}(\frac{|\mathcal{H}_1 \cap \mathcal{S}_t^c|}{|\mathcal{H}_1|} \geq 8\delta) \leq 1/8$. Thus, with probability at least $3/4$, $\frac{|\mathcal{H}_1 \cap \mathcal{S}_t^c|}{|\mathcal{H}_1|} < 8\delta$ and $\frac{|\mathcal{H}_0 \cap \mathcal{S}_t|}{|\mathcal{S}_t|} \leq 8\delta$. To apply the above argument, we just need $8\delta < 1/5$ which holds when $\delta < 1/40$, then $\mathcal{S}_t$ could predict the correct $\pi \in \mathcal{M}_{n,k}$ with probability at least $1/4$.

We will now use an information theoretic inequality to lower bound the $t$ that would make such an estimator possible. Let $P_\pi$ be the probability law of sampling $\tau$ samples from each arm under $\pi$. Then $KL(P_\pi, P_{\pi'}) \leq \Delta^2 \tau |\pi \triangle \pi'|/2 \leq \Delta^2 \tau k$. Directly applying Theorem 2.5 of [30] to our $|\mathcal{M}_{n,k}| \geq (\frac{n}{6k})^{k/4}$ hypotheses, we have that any estimator has a probability of misidentification of at least

$$\frac{1}{2}\left(1 - \frac{2\Delta^2 \tau k}{\log|\mathcal{M}_{n,k}|} - \sqrt{\frac{2\Delta^2 \tau k}{\log^2|\mathcal{M}_{n,k}|}}\right) \geq \frac{1}{2}\left(1 - \frac{8\Delta^2 \tau}{\log(n/6k)} - \sqrt{\frac{32\Delta^2 \tau}{k\log^2(n/k)}}\right)$$

which is at least $1/4$ unless $\tau \gtrsim \Delta^{-2}\log(n/k)$. $\qquad\square$

**Theorem 8** (FWER, FWPD). *Fix $\delta < 3/8$ and $\Delta > 0$. For any $\mathcal{H}_1 \subseteq [n]$ consider an instance $(\{\nu_i\}_{i=1}^n, \mu_0)$ such that $\mu_0 = -\Delta/2$ and $\nu_i = \mathcal{N}(\mu_i, 1)$ where $\mu_i = \Delta/2$ if $i \in \mathcal{H}_1$ and $\mu_i = -\Delta/2$ if $i \in \mathcal{H}_0$. Any algorithm that samples each arm an equal number of times before outputting a set $\mathcal{S} \subseteq [n]$ after $\tau$ total samples, and is FWER-$\delta$ and FWPD-$\delta, \tau$ on $(\{\nu_i\}_{i=1}^n, \mu_0)$ for all $\mathcal{H}_1 \subseteq [n]$ simultaneously, must satisfy $\tau \gtrsim n\Delta^{-2}\log(n)$.*

*Proof.* Fix $t \in \mathbb{N}$. Because the empirical mean is a sufficient statistic for each arm and they are independent, the joint probability distribution of the from the $n$ arms is given by $P_0 := \prod_{i=1}^n \mathcal{N}(\mu_i, 1)$ and the distribution after $t$-pulls on each arm is $P_0^t := \prod_{i=1}^n \mathcal{N}(\mu_i, 1/t)$

For any $j \in [n]$ define $P_j$ as the joint distribution if the $j$th arm's identity was flipped: if $j \in \mathcal{H}_0$ replace its mean with 1, if $j \in \mathcal{H}_1$ replace its means with $\mathcal{H}_0$. Note that $P_j := P_0 \frac{\mathcal{N}(-\mu_i, 1)}{\mathcal{N}(\mu_i, 1)}$ and that $KL(P_j|P_0) = KL(\mathcal{N}(-\mu_i, 1)|\mathcal{N}(\mu_i, 1)) = 2\mu_i^2 = \Delta^2/2$ and $KL(P_j^t|P_0^t) = t\Delta^2/2$.

Now, because the algorithm was assumed FWER-$\delta$ and FWPD-$\delta, t$ on all instances indexed by $\mathcal{H}_1 \subseteq [n]$, it will be able to distinguish between $\{P_k\}_{k=0}^n$ using just $t$ per arm with probability at least $1 - 2\delta \geq 1/4$. The multiple hypothesis testing lower bound of Tsybakov [30, Theorem 2.5], implies that the probability of misclassification of any estimator will be at least

$$\frac{1}{2}(1 - \frac{t\Delta^2}{\log(n)} - \sqrt{\frac{t\Delta^2}{\log^2(n)}}) \geq \frac{1}{4}$$

unless $t \gtrsim \Delta^{-2}\log n$.

$\qquad\square$

## H  Technical Lemmas

**Lemma 8.** *Fix $a \in \mathbb{R}_+^n$ and for $i = 1, \ldots, n$ let $Z_i$ be independent random variables satisfying $\mathbb{P}(Z_i \geq t) \leq \exp(-t/a_i)$. Then*

$$\mathbb{P}\left(\sum_{i=1}^n (Z_i - a_i) \geq t\right) \leq \exp\left(-\min\{\tfrac{t}{4||a||_\infty}, \tfrac{t^2}{4||a||_2^2}\}\right)$$

*and moreover, with probability at least $1 - \delta$, $\sum_{i=1}^n Z_i \leq 5\log(1/\delta)\sum_{i=1}^n a_i$.*

*Proof.* For $i = 1, \ldots, n$ we have that $Z_i = a_i \log(1/\rho_i)$ are independent random variables satisfying $\mathbb{P}(Z_i \geq t) \leq \exp(-t/a_i)$, because the $\rho_i$ are independent *sub-uniformly* distributed random variables. It is straightforward to verify that $\log(\mathbb{E}[\exp(\lambda(Z_i - a_i))]) \leq -a_i\lambda - \log(1 - \lambda a_i) \leq \frac{a_i^2\lambda^2}{2(1 - \lambda a_i)}$ for $\lambda \leq 1/||a||_\infty$. Using the standard Chernoff-bound technique, we have for $\lambda = \min\{\frac{t}{2||a||_2^2}, \frac{1}{2||a||_\infty}\}$ that

$$\begin{aligned}
\mathbb{P}\left(\sum_{i=1}^n (Z_i - a_i) \geq t\right) &\leq \exp\left(-\lambda t + \sum_{i=1}^n \frac{a_i^2\lambda^2}{2(1 - \lambda a_i)}\right)\\
&\leq \exp\left(-\lambda t + \lambda^2||a||_2^2\right)\\
&\leq \exp\left(-\min\{\tfrac{t}{4||a||_\infty}, \tfrac{t^2}{4||a||_2^2}\}\right)\\
&\leq \exp\left(-\tfrac{1}{4}\min\{\tfrac{t}{||a||_1}, \tfrac{t^2}{||a||_1^2}\}\right)
\end{aligned}$$

where the last inequality holds by $||a||_\infty \le ||a||_2 \le ||a||_1 = \sum_{i=1}^n a_i$. The result follows from setting the right hand side equal to $\delta$ and solving for $t$. $\qquad\square$

**Lemma 9.** *Fix* $\delta \in (0, 1/2]$. *Let* $X_i \in [0,1]$ *for* $i = 1, \ldots, m$ *be independent random variables, each satisfying* $\mathbb{P}(X_i \le s) \le s$. *Define* $A(s) = \sum_{i=1}^m \mathbf{1}\{X_i \le s\}$, *then for any* $c \in (0,1)$

$$\mathbb{P}\left(\exists s \in (0,1] : A(s) > sm + (1+4s)\sqrt{2\max\{2s,c\}m\log(\tfrac{\log_2(2/c)}{\delta})} + \tfrac{1+4s}{3}\log(\tfrac{\log_2(2/c)}{\delta})\right) \le \delta.$$

*Moreover,*

$$\mathbb{P}\left(\exists s \in (0,1] : A(s) > sm + (1+2s)\sqrt{4sm\log(\tfrac{2\log_2(2/s)^2}{\delta})} + \tfrac{1+2s}{3}\log(\tfrac{2\log_2(2/s)^2}{\delta})\right) \le \delta.$$

*Also, recall that by the Dvoretzky-Kiefer-Wolfowitz inequality [31] we have*
$$\mathbb{P}\left(\exists s \in (0,1] : A(s) > sm + \sqrt{m\log(1/\delta)/2}\right) \le \delta.$$

*Proof.* First note that $M(s) = \frac{A(s) - \mathbb{E}[A(s)]}{1-s}$ is a martingale with respect to the filtration $\mathcal{F}_t = \{A(s) : s \le t\}$. Thus, for $\lambda > 0$ we have that $\exp(\lambda M(s))$ is a non-negative sub-martingale and we can apply Doob's maximal inequality to obtain

$$
\begin{aligned}
\mathbb{P}\left(\sup_{s\le t} M(s) \ge \epsilon/(1-t)\right) &= \mathbb{P}\left(\sup_{s\le t} \exp(\lambda M(s)) \ge \exp(\lambda\epsilon/(1-t))\right) \\
&\le \exp(-\lambda\epsilon/(1-t))\mathbb{E}\left[\exp(\lambda M(t))\right] \\
&= \exp(-\tfrac{\lambda}{1-t}\epsilon)\mathbb{E}\left[\exp(\tfrac{\lambda}{1-t}(A(t) - \mathbb{E}[A(s)]))\right].
\end{aligned}
$$

Observe that for all $t < 1$ we have

$$\min_\lambda \exp(-\tfrac{\lambda}{1-t}\epsilon)\mathbb{E}\left[\exp(\tfrac{\lambda}{1-t}(A(t) - \mathbb{E}[A(s)]))\right] = \min_\lambda \exp(-\lambda\epsilon)\mathbb{E}\left[\exp(\lambda(A(t) - \mathbb{E}[A(s)]))\right].$$

Noting that $A(t)$ is a sum of independent random variables with each in $[0,1]$ and expectation less than $t$ so that $\mathbb{E}[A(t)] \le mt$, we apply Bernstein's inequality to obtain $\log \mathbb{E}\left[\exp(\lambda(A(t) - \mathbb{E}[A(t)]))\right] \le \frac{mt\lambda^2}{2(1-\lambda/3)}$ for $\lambda \in (0,3)$. Optimizing over $\lambda \in (0,3)$ we have

$$\mathbb{P}\left(\exists s \le t : A(s) > \mathbb{E}[A(s)] + \tfrac{1-s}{1-t}\sqrt{2mt\log(1/\delta)} + \tfrac{1-s}{1-t}\log(1/\delta)/3\right) \le \delta$$

For $k \in \mathbb{N}$ define $T_k = \{s \in [0,1] : 2^{-k-1} < s \le 2^{-k}\}$. Note that $2^{-\lfloor\log_2(2/c)\rfloor} \le c$. So for any $k = 1, 2, \ldots, \lfloor\log_2(2/c)\rfloor$, with probability at least $1 - \frac{\delta}{\lfloor\log_2(2/c)\rfloor}$ we have that for any $s \in T_k$

$$A(s) \le \mathbb{E}[A(s)] + \tfrac{1-s}{1-2^{-k}}\sqrt{2\max\{c, 2^{-k}\}m\log(\log_2(2/c)/\delta)} + \tfrac{1-s}{1-2^{-k}}\log(\log_2(2/c)/\delta)/3.$$

1Note that cases $s \in T_k$ for $k > \lfloor\log_2(2/c)\rfloor$ are handled by $k = \lfloor\log_2(2/c)\rfloor$. For any $k \ge 1$ and $s \in T_k$ we have $s \ge 2^{-k-1}$ so that $1 + 4s \ge \frac{1}{1-\min\{2s, 2^{-1}\}} \ge \frac{1-s}{1-2^{-k}}$ and $2s \ge 2^{-k}$. Thus,

$$\mathbb{P}\left(\exists s \in T_k : A(s) > \mathbb{E}[A(s)] + (1+4s)\sqrt{2\max\{c, 2s\}m\log(\log_2(2/c)/\delta)} + (1+4s)\log(\log_2(2/c)/\delta)/3\right)$$

$$\le \mathbb{P}\left(\exists s \in T_k : A(s) > \mathbb{E}[A(s)] + \tfrac{1-s}{1-2^{-k}}\sqrt{2\max\{c, 2^{-k}\}m\log(\log_2(2/c)/\delta)} + \tfrac{1-s}{1-2^{-k}}\log(\log_2(2/c)/\delta)/3\right)$$

$$\le \delta\frac{1}{\log_2(2/c)} \le \delta\frac{1}{\lfloor\log_2(2/c)\rfloor}.$$

Union bounding over $k = 2, \ldots, \lfloor\log_2(2/c)\rfloor$ handles $\cup_{k=2}^\infty T_k = (0, 1/4]$. To handle $s \in (1/4, 1]$, we note that

$$\mathbb{P}\left(\exists s \in (1/4, 1] : A(s) > \mathbb{E}[A(s)] + (1+4s)\sqrt{2\max\{c, 2s\}m\log(\log_2(2/c)/\delta)} + (1+4s)\log(\log_2(2/c)/\delta)/3\right)$$

$$\le \mathbb{P}\left(\exists s \in T_k : A(s) > \mathbb{E}[A(s)] + \sqrt{m\log(\log_2(2/c)/\delta)}\right)$$

$$\le \left(\delta\frac{1}{\log_2(2/c)}\right)^2 \le \delta\frac{1}{\lfloor\log_2(2/c)\rfloor}$$

where the second to last inequality holds by the DKW inequality [31].

On the other hand, for any $k \geq 1$ and $s \in T_k$ we have $s \geq 2^{-k-1}$ so that $1 + 2s \geq \frac{1-s}{1-\min\{2s,2^{-1}\}} \geq \frac{1-s}{1-2^{-k}}$ and $2^{-k-1} \leq s \leq 2^{-k}$.

$$\mathbb{P}\left(\exists s \in T_k : A(s) > \mathbb{E}[A(s)] + (1+2s)\sqrt{4sm \log(2\log_2(\tfrac{2}{s})^2/\delta)} + (1+2s)\log(2\log_2(\tfrac{2}{s})^2/\delta)/3\right)$$

$$\leq \mathbb{P}\left(\exists s \in T_k : A(s) > \mathbb{E}[A(s)] + \tfrac{1-s}{1-2^{-k}}\sqrt{2 \cdot 2^{-k}m \log(2(k+1)^2/\delta)} + \tfrac{1-s}{1-2^{-k}}\log(2(k+1)^2/\delta)/3\right)$$

$$\leq \delta\frac{1}{2(k+1)^2}.$$

Union bounding over all $k \geq 0$ and noting that $\sum_{k=0}^{\infty} \frac{1}{2(k+1)^2} \leq 1$ completes the proof since $\cup_{k \geq 0} T_k = (0,1]$ $\qquad\qquad\square$

———————————————————————— Recall that the PDF of the $k$-th order statistic of the uniform distribution on $[0,1]$ is given by,

$$\frac{d\mathbb{P}(p_\ell^0 \leq x)}{dx} = \ell\binom{m}{\ell}(1-x)^{m-\ell}x^{\ell-1}.$$

First we consider the case when $\ell > 1$.

$$\sum_{k=1}^{\infty}\frac{1}{k}\mathbb{P}(\frac{(k-1)\alpha}{n} \leq p_\ell^0 \leq \frac{k\alpha}{n}) \leq \mathbb{P}(p_\ell^0 \leq \frac{\alpha}{n}) + \int_{x=\frac{\alpha}{n}}^{1}\frac{\alpha}{nx}d\mathbb{P}(p_\ell^0 \leq x)$$

$$= \mathbb{P}(p_\ell^0 \leq \frac{\alpha}{n}) + \frac{\alpha}{n}\ell\binom{m}{\ell}\int_{x=\frac{\alpha}{n}}^{1}(1-x)^{m-\ell}x^{\ell-2}dx$$

$$\leq \frac{\alpha}{n} + \frac{\alpha}{n}\ell\binom{m}{\ell}(\ell-1)^{-1}\binom{m-1}{\ell-1}^{-1}\int_{\alpha/n}^{1}(\ell-1)\binom{m-1}{\ell-1}(1-x)^{m-\ell}x^{\ell-2}dx$$

$$\leq \frac{\alpha}{n} + \frac{\alpha}{n}\frac{m!}{(\ell-1)!(m-\ell)!}\frac{(\ell-2)!(m-\ell)!}{(m-1)!}$$

$$\leq \frac{\alpha}{n} + \frac{\alpha}{n}\frac{m}{\ell-1}$$

When $\ell = 1$, the argument is slightly different. Firstly note that

$$\mathbb{P}(\frac{(k-1)\alpha}{n} \leq p_\ell^0 \leq \frac{k\alpha}{n}) \leq$$

$$\sum_{k=1}^{\infty}\frac{1}{k}\mathbb{P}(\frac{(k-1)\alpha}{n} \leq p_\ell^0 \leq \frac{k\alpha}{n}) \leq$$