[Reviews · NeurIPS 2018]

Reviewer 1



This paper considers adaptive allocation strategy that guarantees FWER or FDR in n distinct one-armed bandit problems (two-armed bandit problems with parameter \mu_0 known). While I think the results are interesting and the technologies used in this paper are sound (at least with some modifications), the current exposition of paper has many spaces for improvements. Mathematical notions are somewhat loosely used (detailed in major/minor remarks). The contribution of the proposed algorithm is not very easy to follow. On the structure of the paper, (i) 1.3 Related work is rather long. In 8-page submission, a typical length of related work would be < 1 page. (ii) 1.2 summary and 3 main results are overlapping: the first appearance of the main result should be presented with the main theorems. Major remarks >On Table 1 (i) To my understanding, these bounds are upper bounds. Is there any corresponding lower bounds? (ii) What is the number of samples when uniform sampling is adopted? Are improvements of Alg 1 over uniform sampler significant? (iii) Use landau notation O(n \Delta^-2) for upper bound \Omega(n \Delta^-2) for lower bound. >Algorithm 1 Alg 1 is not fully algorithmic because the lines begin with “Apply” are not stated algorithmically (appears to be detailed in Sec 2.1). >On the novelty of Alg 1. The algorithm utilizes standard multiple testing tools (BH and Bonferroni corrections) and the anytime p-value. One fear is that Alg 1 is a naive combination of these technologies. Could the authors highlight the innovative part of the Alg 1? Minor remarks Def 5: smallest time: Is sample complexity random variable? Remark 1 takes me a significant amount of time to understand: I finally got the idea that no algorithm can stop with guarantee of TPR and FWPD, but I believe that the presentation can be improved. Line 97-98 hold only for one-sided test where \mu_i >= \mu_0. Line 133 p-value is undefined at this point Line 211/227 FDR(FWER)-controlled - these properties of sets are undefined. Line 262: independent of all problem parameters: universal constant? Line 265: do not use footnote right after math mode (c^3 looks like c-cubic). Line 267: Do not redefine H_1, H_0, Deltas. Line 271: Neither argument of the minimum follows from the other. > I could not understand. #after author response The author rebuttal is helpful to me as they make some points clear (especially, the lower bounds: line 109-119). The authors state that the sample complexity is a (high-probability-bounded) random variable, which sounds unconventional to me (for example, sample complexities H1 and H2 in "Best Arm Identification in Multi-Armed Bandits" are dist-dependent constants). # Contribution and Presentation As stated in Appendix A, one of the novel contributions of this paper is to avoid union bound over arms to improve logarithmic factors, which I think novel. I found no critical error in the paper. I feel the paper is not very well written in representing their mathematical contents. Some formulas are redefined (e.g. H1, H0, Delta_i, Delta in Thm 1). Lemmas in Appendices are even more sloppy: For example, it takes some efforts to check whether or not P_{i,t} and S_t in pp.11-12 are the same as the ones defined in the main paper. Lemma 2 defines T_i that is used in S_t, which is not defined in the statement of Lemma 2.

Reviewer 2



Summary: This paper studies a bandit approach to sequential experimental design. In particular, the authors consider a sequential game with n arms and the goal is to find arms that have higher means than known (pre-specified) threshold with minimum number of samples. This setting is particularly very interesting for online A/B testing platforms where the platform wishes to detect all the arms (options, offers, etc.) that are better than a given baseline, with lowest number of samples (targeted users) while providing some statistical guarantees. The authors consider two different notions for the False Alarm Control, namely False Discovery Rate (FDR) and Family-Wise Error Rate (FWER) and also two different notions for the Detection Probability, namely True Positive Rate (TPR) and Family-Wise Probability of Detection (FWPD) and provide sample complexity bounds for all 4 pairwise combinations. These complexity bounds are explained in Theorems 1-4. Finally, the authors provide some numerical experiments to demonstrate that the proposed algorithm reduces the sample complexity by a large factor (compared to Successive Elimination and Uniform Allocation rules). Evaluation: The paper is very well-motivated. In the past decade there has been a significant interest in designing adaptive A/B/n testing problems that bring the ideas from multi-armed bandit (MAB) and frequentist statistics together. This paper clearly connects these two ideas. First, it uses UCB-type algorithms to design efficient adaptive allocation rules. Second, it controls the amount of false alarm and detection probability by using the anytime confidence intervals together with BH selection or Bonferroni-like selection. One important aspect of the proposed algorithm is that unlike top k-arm identification algorithms, it does not need to know k in advance. In other words, it can learn the number of arms that have their mean above \mu_0. Furthermore, the papers is nicely written. Authors have done a great job in writing; it is very easy to read and at the same time it is very general that can be modified for using in other settings. I think it would be very nice if the authors could add more simulations to their paper. For instance, it would have been nice to see the number of pulls from different arms in the simulations. Also, simulating some other settings with varying \mu_i (other than the last one) could help with a better understanding of the bounds and the efficiency of the algorithm. Overall, I believe this is a very nice submission and I recommend this paper to be accepted.

Reviewer 3



The paper studies an adaptive measurement allocation problem. The goal is to detect a large number of treatments with positive effect as quick as possible subject to the constraint that number of false discoveries remain small over all time. Two notions of false alarm control are considered: FDR that controls the ratio of false discoveries to all discoveries, and FWER that controls any false discoveries. Also two notions of true discoveries are considered: TPR that requires detecting a large proportion of treatments with positive effect, and FWPD that requires detecting all such treatments. The paper shows sample complexity results for all four combinations of false alarm control and true discoveries. Previous works show such results in the “fixed budget” setting, while this current paper shows guarantees in the “fixed confidence” setting. The paper is well-written and the results seem quite strong. My main comments are as follows: 1) Algorithms designed for the “fixed confidence” setting typically have a stopping condition. As discussed in Remark 1, it is impossible to have a stopping time in your setting. Then how is your algorithm used in practice? When do you decide to terminate the test? 2) The paper needs a more clear explanation of Algorithm 1. 3) A conclusions or future work section is missing.

Reviewer 4



This paper studies an adaptive sampling algorithm for online FDR control. Sample complexity bounds for the same algorithm is provided, where the sample complexity is defined as the smallest time after which an FDR-\delta/FWER-\delta algorithm becomes TPR-\delta or FWPD-\delta. The algorithm is simulated on synthetic data-sets. Overall review: I think the paper is well-written and the theoretical sections are relatively easy to follow. I vote for acceptance but I have some concerns about the experimental evaluation. 1. The second panel of the experiment has |H_1| = 2 << n which is an extreme instance. Then in the third panel |H_1| = O(n). It would be interesting to consider problem instances between these two extremes where |H_1| = n^{\beta} for different values of \beta. The only experiment where \Delta_i's are non-uniform is the last panel and there |H_1| is O(n). I believe experimental results for different intermediate regimes (\beta) should to be provided in which the Delta_i's are non-uniformly (either randomly or deterministically generated). 2. The algorithm should not be denoted as UCB, as this creates confusion w.r.t to the UCB-1 algorithm for K-armed bandits. 3. The algorithm SE should be included in the main body of the paper, as it is not obvious what the successive elimination strategy is in this context. More over there are many algorithms for best arm identification in the fixed confidence setting (https://arxiv.org/abs/1602.04589, https://arxiv.org/abs/1312.7308 etc). I would like to hear the author's thoughts on whether all of them can be adapted for the problem and if so should they be compared with (if implementation is available)? 4. It seems that [https://arxiv.org/pdf/1705.05391.pdf] is very relevant and should be compared with. 5. lines 128-129: "Finally, as a direct consequence of the theoretical guarantees proven in this work .... an algorithm faithful to the theory was implemented and is in use in production at a leading A/B testing platform." It would be great if the authors include the results of these experiments (at least qualitatively) in the paper, rather than just stating that such an algorithm was implemented.